# Rho1 activation recapitulates early gastrulation events in the ventral, but not dorsal, epithelium of *Drosophila* embryos

Ashley Rich, Richard G Fehon, Michael Glotzer*

Department of Molecular Genetics and Cell Biology, University of Chicago, Chicago, United States

**Abstract** Ventral furrow formation, the first step in *Drosophila* gastrulation, is a well-studied example of tissue morphogenesis. Rho1 is highly active in a subset of ventral cells and is required for this morphogenetic event. However, it is unclear whether spatially patterned Rho1 activity alone is sufficient to recapitulate all aspects of this morphogenetic event, including anisotropic apical constriction and coordinated cell movements. Here, using an optogenetic probe that rapidly and robustly activates Rho1 in *Drosophila* tissues, we show that Rho1 activity induces ectopic deformations in the dorsal and ventral epithelia of *Drosophila* embryos. These perturbations reveal substantial differences in how ventral and dorsal cells, both within and outside the zone of Rho1 activation, respond to spatially and temporally identical patterns of Rho1 activation. Our results demonstrate that an asymmetric zone of Rho1 activity is not sufficient to recapitulate ventral furrow formation and reveal that additional, ventral-specific factors contribute to the cell- and tissue-level behaviors that emerge during ventral furrow formation.

## Introduction

Tissue morphogenesis underlies the development of multicellular organisms. The molecular and cellular mechanisms that govern tissue morphogenesis remain a central challenge in developmental cell biology. Extensive genetic and biochemical experiments have defined the factors required for many morphogenetic movements. Furthermore, methods for imaging and quantitatively describing cell shape changes are ever-improving. Despite this progress, questions remain. For example, how pliable are tissues before and while they are deforming? To what degree do existing cellular structures limit the ability of a tissue to deform, and to what extent and by what mechanism are shape changes of neighboring cells coordinated?

Ventral furrow formation in the *Drosophila* embryo is one of the best studied examples of tissue morphogenesis; it is the first step in *Drosophila* gastrulation. Ventral furrow formation occurs when a rectangular zone of approximately 1000 cells, arranged in 18 rows, on the ventral surface of the embryonic epithelium apically constrict and invaginate into the embryo, ultimately giving rise to the embryonic mesoderm (*Leptin and Grunewald, 1990*; *Sweeton et al., 1991*). Many molecules required for ventral furrow formation have been identified: an extracellular serine protease cascade activates the transcription factor Dorsal which drives the expression of two additional transcription factors, Snail and Twist, in a subset of ventral cells, inducing them to adopt mesodermal fates (*Morisato and Anderson, 1995*; *Ip et al., 1992*; *Jiang et al., 1991*). Snail and Twist then induce the expression of secreted and cell surface molecules, including the ligand Fog, the G-protein-coupled receptor (GPCR) Mist, and the transmembrane protein T48 (*Dawes-Hoang et al., 2005*; *Costa et al., 1994*; *Kölsch et al., 2007*; *Manning et al., 2013*). Together with Concertina, a maternally contributed Gα protein, and Smog, a maternally contributed GPCR, these factors recruit and activate RhoGEF2, a Rho1-specific guanine nucleotide exchange factor, at the apical membrane of

*For correspondence:
mglotzer@uchicago.edu

Competing interests: The authors declare that no competing interests exist.

ventral cells (*Parks and Wieschaus, 1991*; *Kölsch et al., 2007*; *Nikolaidou and Barrett, 2004*; *Kerridge et al., 2016*). RhoGEF2 then activates Rho1 to assemble a contractile actomyosin network (*Martin et al., 2009*; *Fox and Peifer, 2007*); these networks within single cells are coupled through adherens junctions between neighboring cells into a supracellular actomyosin network that promotes robust ventral furrow formation (*Martin et al., 2010*; *Yevick et al., 2019*). Notably, both RhoGEF2 accumulation and Rho1 activation are pulsatile (*Martin et al., 2010*; *Mason et al., 2016*).

The intracellular signaling cascade described above activates Rho1 within individual presumptive mesoderm cells. This could, in principle, account for ventral furrow formation (*Gilmour et al., 2017*; *Ko and Martin, 2020*). However, several features of the ventral furrow suggest that ventral cells exhibit a high degree of intercellular coupling, which may influence the outcome of the genetically encoded contractility. For example, ventral cell apical constriction is anisotropic, occurring more along the dorsal-ventral than the anterior-posterior axis of the embryo (*Sweeton et al., 1991*; *Martin et al., 2010*). If individual ventral cells constrict and invaginate without being influenced by their neighbors, one would predict isotropic apical constriction. Additionally, the apical constriction of individual cells appears coordinated, with cells adjacent to constricting cells more likely to constrict than their more distant counterparts (*Sweeton et al., 1991*; *Gao et al., 2016*). Furthermore, multiple rows of cells lateral to the furrow bend toward it, indicating that forces are transmitted over long distances in the ventral epithelium (*Rauzi et al., 2015*; *Costa et al., 1994*; *Leptin et al., 1992*).

Taken together, this wealth of previous results suggests that ventral furrow formation results from a combination of intracellular Rho1-mediated contractility and intercellular coupling of those contractile forces. In the simplest iteration of this model, an asymmetric zone of Rho1 activation is sufficient to recapitulate both the intra- and intercellular aspects of ventral furrow formation (*Doubrovinski et al., 2018*). Indeed, it was recently shown that an asymmetric zone of local Rho1 activation is sufficient to induce an ectopic furrow in the dorsal *Drosophila* epithelium (*Izquierdo et al., 2018*). However, it remains unclear whether local Rho1 activation alone is sufficient to induce sustained tissue morphogenesis and recapitulate all aspects of ventral furrow formation. Alternatively, ventral-specific gene expression may endow ventral cells with factors that collaborate with active Rho1 to generate the observed cell shape changes. If this is the case, an asymmetric zone of Rho1 activity would yield distinct cell shape changes in the dorsal versus the ventral embryonic epithelium.

Addressing these and related questions necessitates the ability to activate Rho1 with high spatial and temporal precision without otherwise perturbing the embryo. Optogenetic techniques utilize photosensitive proteins to control protein localization and/or activity with light; these techniques are, therefore, well-suited to interrogate the basis for the anisotropic and coordinated nature of apical constriction during ventral furrow formation. Importantly, the ideal optogenetic approach will activate Rho1 in response to light alone.

Here, we use a LOV-domain-based optogenetic probe to acutely activate Rho1 in *Drosophila*. We demonstrate that this system expresses ubiquitously throughout *Drosophila* development and is well tolerated. Optogenetic activation of Rho1 induces ectopic deformations in both the dorsal and ventral embryonic epithelium during early embryogenesis. We find that ventral, but not dorsal, cells respond to ectopic Rho1 activation with aligned, anisotropic apical constriction. This ventral-specific response requires Dorsal and Twist expression. Furthermore, we provide evidence that the transmission of contractile forces over long distances is specific to the ventral epithelium.

## Results

### A LOV-domain-based optogenetic system controls Rho1 activity in *Drosophila*

To study the cellular consequences of acute Rho1 activation and probe the impact of Rho1 activation on cells neighboring the activation region, we adapted an optogenetic system for use in *Drosophila*. This two-component system consists of a membrane tethered LOV domain fused to the SsrA peptide and a cytoplasmic SspB protein fused to a protein of interest (*Figure 1a*; *Guntas et al., 2015*; *Strickland et al., 2012*). Blue light exposure induces a conformational change in the LOV domain, exposing the SsrA peptide and recruiting the SspB fusion protein to the plasma membrane (*Figure 1a*). As a proof of concept, we first expressed the membrane localized LOV domain and an

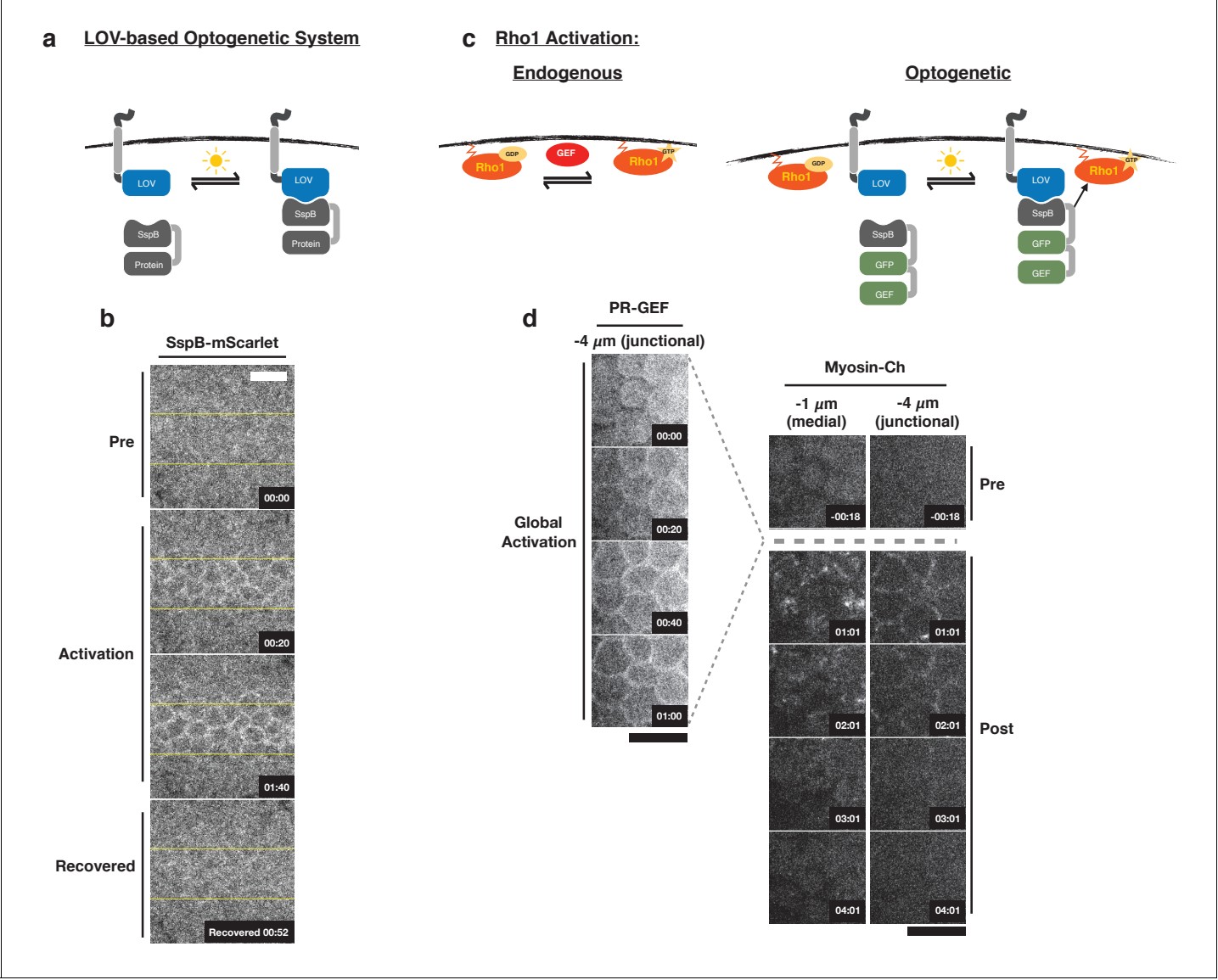

**Figure 1.** Optogenetic control of Rho1 in *Drosophila*. (a) Generic LOV domain-based optogenetic system consisting of a membrane localized LOVSsrA protein and a recruitable SspB protein. SspB can be fused to any protein of interest. Blue light induces a conformational change in the LOV domain, allowing it to recruit SspB fusion proteins to the membrane. (b) Dorsal epithelium of an embryo expressing the membrane-localized LOVSsrA and SspB-mScarlet just after the completion of cellularization before, during, and after photoactivation in the indicated region (yellow box). Data representative of 4/4 embryos. (c) Optogenetic system for activating Rho1: SspB is fused to GFP and the Dbl homology (DH) domain of LARG (PR-GEF). Photoactivation induces recruitment of the PR-GEF to the membrane and Rho1 activation. (d) Dorsal epithelium of an embryo expressing a membrane-localized LOVSsrA and PR-GEF just after the completion of cellularization. The distribution of PR-GEF and myosin are shown before, during, and at the indicated times after global photoactivation. Data representative of 5/5 embryos. Note that the light dose used for global experiments is higher than the light dose used for local activation experiments. Time zero indicates the beginning of photoactivation. Scale bars are 10 μm.

The online version of this article includes the following figure supplement(s) for figure 1:

**Figure supplement 1.** Recruitment of PR-GEF activates Rho1 in all tissues tested.
**Figure supplement 2.** Ectopic Rho1 activation is sensitive to light dose.
**Figure supplement 3.** Inactivation kinetics of the LOV domain dictate the off rate of optogenetic-induced Rho1 activity.

SspB-mScarlet fusion from the ubiquitin promoter. Local activation of a region of the dorsal embryonic epithelium with blue light induces rapid recruitment of SspB-mScarlet to the plasma membrane (*Figure 1b*). SspB-mScarlet remains associated with the plasma membrane as long as blue light

activation is sustained but rapidly (~1 min), upon cessation of photoactivation, returns to its dark state, diffusely distributed in the cytoplasm (*Figure 1b*).

To control Rho1 activation, we replaced mScarlet with the catalytic Dbl homology (DH) domain of LARG to generate photorecruitable SspB-GFP-LARG(DH) (hereafter called PR-GEF) (*Figure 1c*). LARG is a human RhoA-specific GEF; the DH domain of LARG has previously been used in a related optogenetic system to control RhoA activity in mammalian tissue culture cells (*Wagner and Glotzer, 2016*). We used the DH domain of LARG alone to ensure that the recruitable GEF's function is divorced from all endogenous regulation and only sensitive to optogenetic activation. Homozygous flies expressing the membrane localized LOVSsrA and PR-GEF from the ubiquitin promoter are viable and fertile, indicating that these transgenes are well tolerated. Global activation of the dorsal embryonic epithelium with blue light induces strong recruitment of PR-GEF to the plasma membrane within seconds, and this global PR-GEF recruitment induces cortical myosin accumulation within 1 min (*Figure 1d*). Myosin accumulates apically, both medially and junctionally (*Figure 1d*). Optogenetically induced cortical myosin completely disappears within 3 min of cessation of photoactivation (*Figure 1d*). Thus, using conventional microscopy, this optogenetic system rapidly, robustly, and reversibly activates Rho1 in the embryonic epithelium. This system also activates Rho1 in all *Drosophila* tissues tested, including the pupal notum, follicular epithelium, larval wing imaginal disc, and larval central nervous system (CNS) (*Figure 1—figure supplement 1*). In the larval wing periodal epithelium, optogenetic activation of Rho1 can induce myosin accumulation with subcellular precision (*Figure 1—figure supplement 1c*). Optogenetic activation of Rho1 is sensitive to light dosage; attenuating the activating light induced less myosin-Ch accumulation, indicating lower levels of Rho1 activation (*Figure 1—figure supplement 2*). Above a certain threshold of activating light, Rho1 becomes globally activated, despite precisely defined activation regions (*Figure 1—figure supplement 2*). Thus, this LOV domain-based optogenetic probe is capable of controlling Rho1 activation with high spatial and temporal resolution. Furthermore, the level of Rho1 activation can be tuned by modulating light dosage.

While this LOV domain-based optogenetic probe recovers to its dark state activity level within minutes (*Figure 1*), some biological phenomena may require faster recovery kinetics. To increase the inactivation rate of our optogenetic probe, we introduced a previously identified point mutation, I427V, into the LOV domain, which increases the rate at which the LOV domain returns to the dark state (*Christie et al., 2007*). I427V increases the inactivation rate of the optogenetic system in *Drosophila* S2 cells expressing a membrane localized LOV domain containing this mutation and a cytoplasmic, recruitable tagRFP-SspB (*Figure 1—figure supplement 3a,b*). Increasing the recovery rate of the LOV domain also decreases the maximum recruitment of tagRFP-SspB (*Figure 1—figure supplement 3a,b*), demonstrating the trade off between rapid inactivation and total recruitment. Global activation of the rapid cycling LOV domain in *Drosophila* larval brains induced robust Rho1 activation, as scored by accumulation of a Rho1 biosensor (*Figure 1—figure supplement 3c*); this Rho1 activity dissipated within a minute of cessation of global optogenetic activation. In contrast, Rho1 remained active a minute after global activation of the wild-type LOV domain (*Figure 1—figure supplement 3c*). Thus, the cycling kinetics of the LOV domain are the primary determinant of the off rate of optogenetic-induced Rho1 activity. This emphasizes that there are rapid and robust mechanisms for shutting off Rho1 activity in vivo; furthermore, it suggests that cells continually activate Rho1 during cellular and developmental processes that require sustained Rho1 activation. The wild-type LOV domain is used for the remainder of the experiments presented, as the rapid recovery was not essential to address the questions answered here.

## Quantitative baseline of ventral furrow formation in wildtype embryos

To establish a baseline for comparison of subsequent perturbation experiments, we quantitatively analyzed the progression of the endogenous ventral furrow in embryos expressing the optogenetic components and the membrane marker Gap43-mCherry (*Figure 2a*). We acquired time-lapse images of non-activated embryos and subsequently segmented tissues, tracked individual cells, and quantified the area of the apical-most surface of ventral cells at 5 min intervals (*Aigouy et al., 2010*). We quantified the anisotropy of the apical-most surfaces of cells at these same time intervals (*Aigouy et al., 2010*). Ventral cells in these embryos constrict their apical surfaces over the course of 10–15 min (*Figure 2,* 10-20 min time points). This timing is consistent with previously published work, suggesting the optogenetic probe does not impair ventral furrow formation (*Rauzi et al.,*

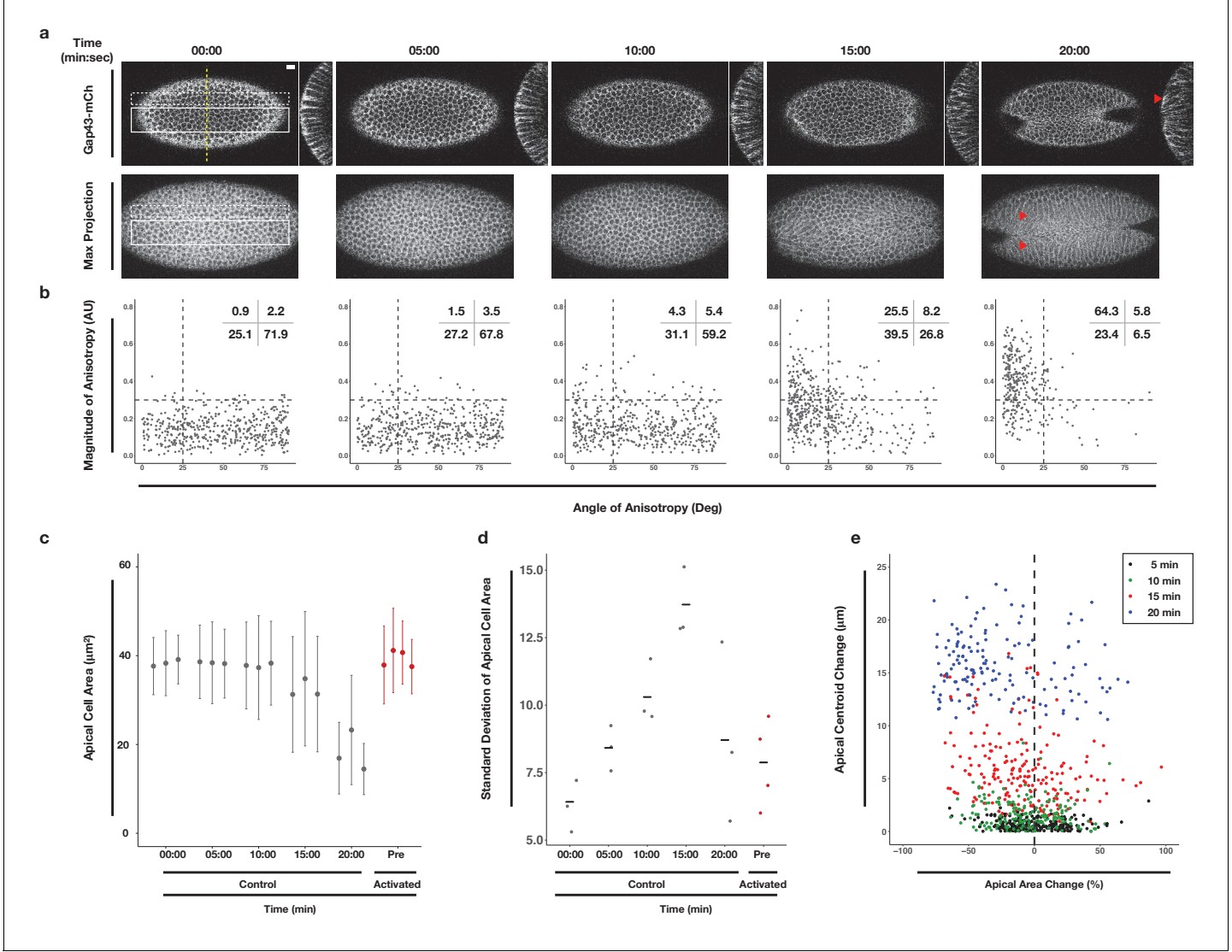

**Figure 2.** Morphometric analysis of ventral cells during ventral furrow formation. (a) Ventral epithelium of a *Drosophila* embryo expressing the optogenetic components and Gap43-Ch but not exposed to activating light before, during, and after the onset of ventral furrow formation. Single planes (top), YZ projections (side), and maximum intensity projections (bottom) are shown. Scale bar is 10 μm. Filled arrowheads indicates long-range bending of these cells. Time zero indicates start of filming. (b) Anisotropy scatter plots: Each dot represents a single cell within the solid white box in (a) at the corresponding time. The magnitude of anisotropy is plotted on the y-axis; the orientation of anisotropy, relative to the anterior-posterior axis of the embryo, is plotted on the x-axis. Dotted lines are provided to facilitate comparisons between plots. Insets show percentage of cells in each quadrant. Cells in the upper left quadrant exhibit highly aligned, anisotropic apical constriction. (c) Quantification of apical cell area for cells within solid white box in (a). Gray columns represent unactivated cells. Red columns represent optogenetically activated cells discussed in *Figure 4*. (d) Standard deviation of the apical cell areas shown in (c). Gray and red dots represent standard deviation of apical cell areas for each non-activated (gray) or activated (red, see *Figure 4*) embryo at time point indicated along the X axis. Black crossbars indicate the average standard deviation for a given time point. (e) Quantification of the bending of cells neighboring the ventral midline (white dotted box in a). X axis is the percent apical area change relative to 00:00 min; Y axis is the change in position of the centroid of the apical cell surface along the dorsal-ventral axis relative to 00:00 min. Data in b-d are from 463 cells from three embryos. Data in e are from 203 cells from three embryos.

The online version of this article includes the following figure supplement(s) for figure 2:

**Figure supplement 1.** Neither magnitude nor alignment of anisotropy correlates with extent of apical constriction.

**Figure supplement 2.** Apical constriction of ventral cells occurs just after the onset of myosin accumulation.

*2015*). Notably, the variance of apical cell areas increases as ventral furrow formation begins (*Figure 2d*). This reflects the fact that ventral cells do not constrict in unison, but rather in a salt-and-pepper fashion (*Sweeton et al., 1991*; *Gao et al., 2016*). The apical constriction of ventral cells is

anisotropic, occurring much more along the dorsal-ventral axis than the anterior-posterior axis (*Figure 2b*). The magnitude and alignment of this anisotropy increases over time (*Figure 2b*), but anisotropy does not correlate with the extent of apical constriction (*Figure 2—figure supplement 1*). Anisotropic, apical constrictions begin within 2 min of the first detectable myosin accumulation in ventral cells (*Figure 2—figure supplement 2*). Finally, we quantified the movement of the apical surfaces of cells neighboring the ventral midline. Specifically, we tracked the centroids of these cells' apical surface over time along the dorsal-ventral axis. The apical surfaces of cells that neighbor the central cells of the ventral furrow move substantially along the dorsal-ventral axis during ventral furrowing, reflecting their bending toward the invaginating furrow (*Figure 2a,e*). These data confirm that our optogenetic probes are minimally perturbing and provide a standardized timeline along which cell shape changes occur during ventral furrow formation in this genetic background.

## Rho1 activation is sufficient to induce reversible furrows in the *Drosophila* embryonic epithelium

Having validated that this optogenetic system reversibly activates Rho1 in the early *Drosophila* embryo and that the expression of the probe does not adversely impact endogenous ventral furrow formation, we asked whether an asymmetric zone of Rho1 activation is sufficient to induce an ectopic invagination in the embryonic dorsal epithelium of embryos expressing the optogenetic components, Myosin-Ch, and nearing the end of cellularization. Twelve pulses of local photoactivation of Rho1 every 20 s was sufficient to induce apical myosin accumulation and an ectopic furrow in the dorsal epithelium (*Figure 3a*). Importantly, the size of the furrowed region closely mirrors the size of the photoactivated zone, demonstrating the spatial precision of this approach and emphasizing that this deformation is light-dependent. This is consistent with recently published work (*Izquierdo et al., 2018*). Except where explicitly stated, Rho1 is activated in asymmetric zones of the same dimensions throughout this work.

Local Rho1 activation also induces myosin accumulation and ectopic furrows in the ventral embryonic epithelium prior to the invagination of the endogenous ventral furrow (*Figure 3b*). Several lines of evidence argue that the tissue-level shape changes and myosin accumulation observed in the ventral epithelium are light-dependent. First, these optogenetic experiments were performed in embryos with little to no detectable myosin accumulation in the ventral epithelium. Furthermore, the optogenetically induced myosin accumulation in the ventral epithelium often dissipates before myosin accumulates in the endogenous ventral furrow, and, in some cases, the optogenetic furrow is offset from the eventual ventral midline, as seen in *Figure 3b*. Additionally, when Rho1 is optogenetically activated in a rectangular zone, perpendicular to the anterior-posterior axis of the embryo and the eventual ventral furrow, ectopic Myosin-Ch accumulation is induced both prior to and following the onset of endogenous ventral furrow formation (*Figure 3—figure supplement 1*). Even when low levels of endogenous myosin are present in the presumptive ventral furrow, the majority of myosin-Ch accumulation within the photoactivated region is light-dependent, as it accumulates within the defined region and recovers to the pre-activation state following the cessation of photoactivation (*Figure 3—figure supplement 1*). Notably, ectopic myosin accumulation is restricted to the defined activation zone, suggesting that optogenetic activation of Rho1 in the ventral epithelium does not induce detectable non-cell autonomous myosin accumulation.

The ectopic furrows induced in either the dorsal or ventral epithelium recover to their pre-activation state within 4 min of cessation of optogenetic Rho1 activation (*Figure 3b*; *Figure 3—figure supplement 2*). These recovery kinetics are slightly longer than the cycling kinetics of the WT LOV domain (*Figure 1d*, *Figure 1—figure supplement 3*). Thus, while ectopic Rho1 activation induces ectopic deformations, the downstream consequences of optogenetic Rho1 activation are rapidly and robustly inactivated in the absence of continued photoactivation.

To characterize the molecular nature of these optogenetically induced furrows, we assayed myosin accumulation along the apicobasal axis of dorsal or ventral cells following optogenetic activation of Rho1. Toward this end, we activated Rho1 in embryos depleted of RhoGEF2 to specifically assay optogenetically induced myosin accumulation. Photoactivation-induced Myosin-Ch accumulation is restricted to the apical-most 3 or 4 µm of activated dorsal and ventral cells, and similar levels of myosin accumulation are induced in dorsal and ventral epithelia (*Figure 3—figure supplement 3*). This restricted accumulation of myosin along the apicobasal axis of activated cells suggest that lateral

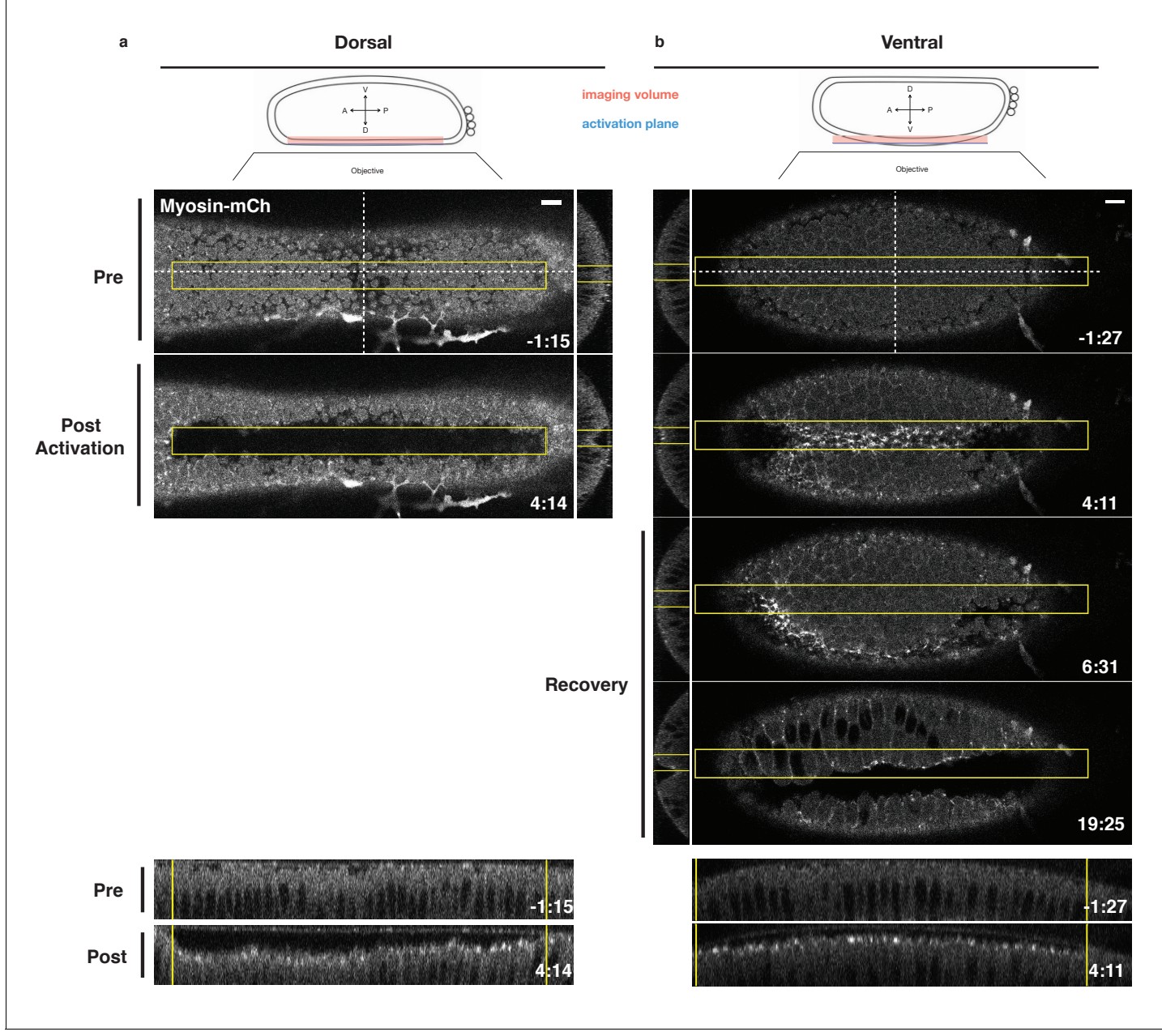

**Figure 3.** Local Rho1 activation is sufficient to induce ectopic furrows in the embryonic epithelium. (a,b) Top: Schematic of local activation set up. Bottom: Embryos expressing the optogenetic components and Myosin-Ch prior to the invagination of the endogenous ventral furrow were optogenetically activated in a single apical plane within the yellow box. Ectopic furrows and Myosin-Ch accumulation are shown in the dorsal (a) and ventral (b) epithelium. XZ projections shown in middle, YZ projections shown at bottom. Optogenetic activation of Rho1-induced ectopic furrows in 7/7 dorsal (a) and 5/5 ventral (b) epithelia. Optogenetically-induced furrows recovered in 4/4 ventral epithelia (b). Time zero indicates the first pulse of blue light activation. Scale bars are 10 μm.

The online version of this article includes the following figure supplement(s) for figure 3:

**Figure supplement 1.** Optogenetic Rho1 activation induces Myosin-Ch accumulation in the ventral epitheium that does not spread outside of the activation region.

**Figure supplement 2.** Optogenetically induced furrows in dorsal epithelium revert following cessation of Rho1 activation.

**Figure supplement 3.** Optogenetic Rho1 activation induces apical myosin accumulation in both dorsal and ventral epithelia.

shortening along the length of the apicobasal axis is not a major contributor to the ectopic furrows induced by PR-GEF recruitment.

## Rho1 activation induces distinct apical constriction in the dorsal and ventral epithelium

The previous results demonstrate that asymmetric zones of Rho1 activity are sufficient to induce ectopic furrows in both dorsal and ventral epithelia. To determine whether dorsal and ventral cells within these light-induced furrows respond similarly to spatially and temporally identical zones of Rho1 activation, we locally activated Rho1 in the dorsal or ventral epithelium of embryos expressing the optogenetic components and the membrane marker Gap43-mCh just before the completion of cellularization and analyzed the subsequent cells shape changes as in *Figure 2*. Local Rho1 activation in individual cells or in a group of cells in the dorsal embryonic epithelium induced apical constriction (*Figure 4a–c*). Unlike cells of the endogenous ventral furrow, which exhibit strongly anisotropic apical constriction (*Figure 2b*), optogenetically activated dorsal cells constrict isotropically (*Figure 4d* - bottom panel left, *Figure 4—figure supplement 1*, *Figure 4—figure supplement 2*). Thus, an asymmetric, rectangular zone of Rho1 activation via this probe is not sufficient to fully recapitulate the cell shape changes associated with endogenous ventral furrowing. This result differs from previous work (See Discussion) (*Izquierdo et al., 2018*).

We next investigated whether optogenetic Rho1 activation has a different effect on ventral cells, which express ventral-specific genes. Embryos for these photoactivation experiments in the ventral epithelium were chosen based on several criteria to ensure that the ventral cells were photoactivated prior to the onset of endogenous ventral furrow formation. First, embryos were selected to be in the final stages of cellularization, before they exhibited any overt signs of apical constriction. Post-experiment analysis of pre-activation cell areas confirmed that these experiments were performed prior to the onset of cell shape changes and endogenous ventral furrow formation (*Figure 2c*, compare red column with gray columns). Furthermore, the standard deviation of the areas of activated cells in each embryo prior to photoactivation further indicate that these ventral cells are photoactivated prior to the onset of endogenous ventral furrow formation (*Figure 2d*, compare red column with gray columns). Importantly, myosin accumulation immediately precedes cell shape changes (*Figure 2—figure supplement 2*), suggesting that these optogenetic experiments begin before endogenous myosin accumulation commences.

Optogenetic activation of Rho1 in a collection of ventral cells also induced apical constriction within the zone of Rho1 activation (*Figure 4b,c*). In contrast to dorsal cells, the apical surfaces of activated ventral cells were elongated, indicating anisotropic apical constriction. This anisotropy was strongly aligned with the anterior-posterior axis (*Figure 4d* -bottom right panel, *Figure 4—figure supplement 1*, *Figure 4—figure supplement 2*), but, as in non-activated ventral tissue, this anisotropy does not correlate with amount of apical constriction (*Figure 2—figure supplement 1*). This anisotropic apical constriction of ventral cells (*Figure 4d*) is induced by a 4-min activation protocol and is visible within the first 2 min of photoactivation. Given that these cells were photoactivated prior to the onset of ventral furrow formation (*Figure 2b,c*), this level of anisotropy is precocious relative to the endogenous ventral furrow. Furthermore, a smaller, asymmetric zone of Rho1 activation also induces anisotropic apical constriction in activated ventral cells but not their non-activated neighbors (*Figure 4—figure supplement 3*), confirming that our optogenetic experiments recapitulate the cell shape changes seen during endogenous ventral furrow but induce these cell shape changes to occur earlier and faster than they normally would. Thus, in contrast to the isotropic apical constriction induced in activated cells within the dorsal epithelium, optogenetic Rho1 activation in cells within the ventral epithelium induces precocious, anisotropic apical constriction that strongly resembles the anisotropic apical constrictions seen during endogenous ventral furrow formation (*Figure 2b*). Importantly, our analysis of myosin accumulation demonstrates that the optogenetic input of Rho1 activity is similar in both dorsal and ventral epithelia (*Figure 3—figure supplement 3*), ruling out the possibility that the differential behavior of dorsal and ventral cells is due to the optogenetic probes behaving distinctly in the two epithelia.

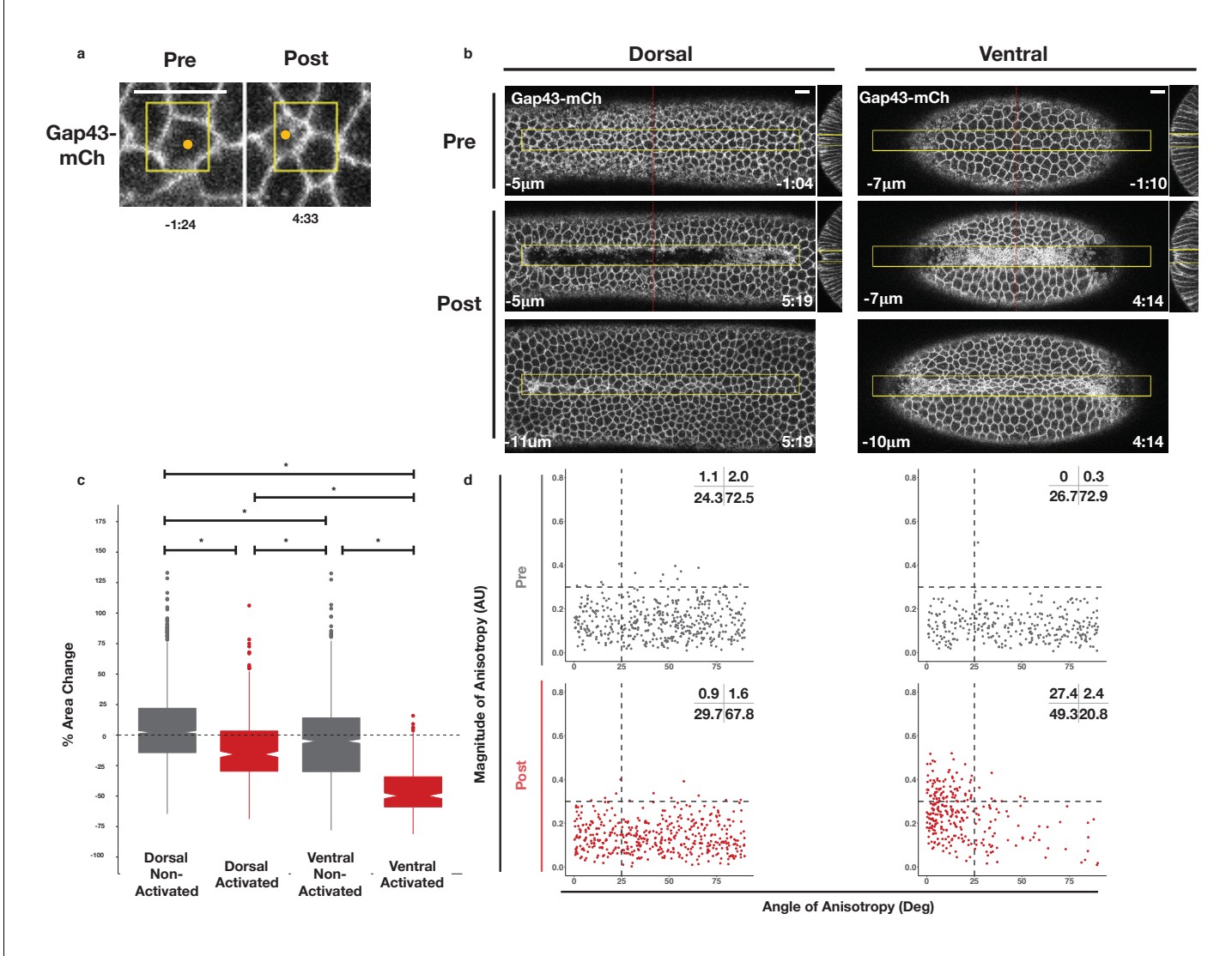

**Figure 4.** Rho1 activation induces distinct apical constriction in dorsal and ventral epithelial cells. (a) Dorsal cells from an embryo expressing the optogenetic components and Gap43-mCh before the onset of cell shape changes before and after photoactivation within the yellow box. Z-slices shown represent the apical-most view of the activated cell (orange dot). Data representative of 7/8 cells from two embryos. (b) Dorsal (left) and ventral (right) epithelium of embryos expressing the optogenetic components and Gap43-mCh before the onset of cell shape changes. Rho1 was activated in the yellow box. Images shown in bottom panels were chosen to show the apical surfaces of activated cells. Red lines in (b) indicate position of each YZ slice. Data are representative of 4/4 (a) and 4/4 (b) embryos. Time zero indicates the first pulse of blue light activation. Scale bars are 10 μm. (c) Quantification of apical area change induced by optogenetic Rho1 activation. Gray columns represent non-activated cells (cells outside the yellow box in (b)). Red columns represent activated cells (cells within the yellow box in (b)). Statistical Analysis: Kruskal-Wallis test, Chi$^2$ = 691.6, df = 3, p = 2.2 x 10$^{-16}$; pairwise comparisons with Wilcoxon signed rank text, using BH method to adjust for multiple comparisons. * indicates p < 0.05. (d) Anisotropy scatter plots as in *Figure 2b*: Each dot represents an activated dorsal (left) or ventral (right) cell before (top) or after (bottom) optogenetic activation of Rho1. A total of 444 dorsal cells from four embryos and 288 ventral cells from four embryos were analyzed. See *Figure 4—figure supplement 1* for plots of the changes in anisotropy of individual cells. See *Figure 4—figure supplement 2* for statistical analysis of data in d.

The online version of this article includes the following figure supplement(s) for figure 4:

**Figure supplement 1.** WT ventral, but not dorsal, cells exhibit large changes in the magnitude and alignment of anisotropy in response to Rho1 activation.

**Figure supplement 2.** Comparisons of angle and magnitude of anisotropy before and after optogenetic Rho1 activation.

**Figure supplement 3.** Optogenetic activation of Rho1 induces precocious cell shape changes in the ventral epithelium.

**Figure supplement 4.** Schematic of data collection and analysis for local activation experiments.

## Genetic requirements for ventral-specific responses

The finding that an asymmetric zone of Rho1 activation induces differential responses in the dorsal and ventral epithelia suggests that ventral patterning may influence the response to Rho1 activation. Ventral-specific factors, such as those downstream of Dorsal, Twist, and/or Snail (*Figure 5a*), may cooperate with an asymmetric zone of Rho1 activation during endogenous ventral furrow formation to drive strong, anisotropic apical constriction. To test this hypothesis, we locally activated Rho1 in ventral cells lacking Dorsal protein, a factor required for ventral identity. Optogenetic Rho1 activation in embryos derived from females homozygous for a null *dorsal* allele still induced apical constriction, but these apical constrictions were weaker than those of WT ventral cells and were no longer anisotropic (*Figure 5b and c*, *Figure 5—figure supplement 1*, *Figure 4—figure supplement 1*, *Figure 4—figure supplement 2*). Indeed, in the absence of the Dorsal protein, the response of ventral cells to Rho1 activation is similar to the response of wildtype cells in the dorsal epithelium (*Figure 5c* vs. *Figure 4d* -Activated Dorsal, *Figure 4—figure supplement 1*, *Figure 4—figure supplement 2*). Thus, Dorsal is required to predispose ventral cells to constrict anisotropically along the anterior-posterior axis of the embryo.

The transcription factor Twist is downstream of Dorsal activity in ventral cells. We optogenetically activated Rho1 in ventral cells of embryos homozygous for a null allele of *twist*. These mutant cells exhibited apical constriction following ectopic Rho1 activation (*Figure 5b*, *Figure 5—figure supplement 1*), but the amount of constriction, magnitude of anisotropy, and the degree of alignment with the anterior-posterior axis was less than that of wildtype ventral cells (*Figure 5c* v. *Figure 4d* -Activated Ventral ; *Figure 4—figure supplement 1*, *Figure 4—figure supplement 2*). Thus, Twist promotes the predisposition of ventral cells to constrict anisotropically along the anterior-posterior axis of the embryo. This suggests Twist plays a Rho1-independent role that influences the cell shape changes executed by cells of the ventral furrow.

We note that ventral cells lacking Twist exhibit more aligned apical constriction than ventral cells lacking Dorsal (*Figure 5c*, *Figure 4—figure supplement 1*, *Figure 4—figure supplement 2*). These distinct responses suggest that there is a Twist-independent mechanism downstream of Dorsal that promotes aligned anisotropic apical constriction in response to Rho1 activation. We speculate that Snail is responsible for this Twist-independent behavior, but repeated attempts to combine a null *snail* allele with our optogenetic components failed, so we were not able to test this hypothesis.

Dorsal is required for and Twist promotes ventral cells to respond to ectopic Rho1 activation with strong, aligned anisotropic apical constriction. This may reflect that the transcriptional targets of Dorsal and Twist are required for Rho1 activation during endogenous ventral furrow formation (*Martin et al., 2009*). Thus, to determine whether Dorsal and Twist contribute to anisotropic apical constriction by regulating the expression of Rho1 activators or by regulating the expression of factors that cooperate with Rho1 activators, we optogenetically activated Rho1 in ventral cells depleted of RhoGEF2, the endogenous activator of actomyosin contractility during ventral furrow formation. RhoGEF2 is required for proper organization of the actomyosin cytoskeleton during cellularization, and embryos lacking RhoGEF2 have some cellularization defects, contributing to irregularities in the epithelium (*Padash Barmchi et al., 2005*). Thus, prior to optogenetic activation, a subset of cells depleted of RhoGEF2 are anisotropic, though randomly aligned (*Figure 5c*). Despite the non-uniformity in these epithelia, optogenetic activation of Rho1 in ventral cells depleted of RhoGEF2 increased the extent of aligned, anisotropic apical constriction (*Figure 5c*, *Figure 5—figure supplement 1*, *Figure 4—figure supplement 1*, *Figure 4—figure supplement 2*). Optogenetic activation of Rho1 in dorsal cells depleted of RhoGEF2 does not induce the same amount of highly aligned and anisotropic apical constriction (*Figure 4—figure supplement 1*, *Figure 4—figure supplement 2*). These results suggest ventral cells can respond to optogenetic Rho1 activation with anisotropic apical constriction even when the level of endogenous Rho1 activity is greatly attenuated. However, elevated Rho1 levels may contribute to strong, aligned, anisotropic apical constriction during endogenous ventral furrow formation.

Interestingly, in contrast to the rapid reversion of ectopic deformations in wild-type tissues, ectopic furrows induced in the ventral or dorsal epithelium of embryos lacking RhoGEF2 failed to revert following cessation of optogenetic activation. This suggests Rho1 activity downstream of RhoGEF2 makes a significant contribution to the tension that provides a restorative force throughout these epithelia, likely through its role in organizing the actomyosin cytoskeleton.

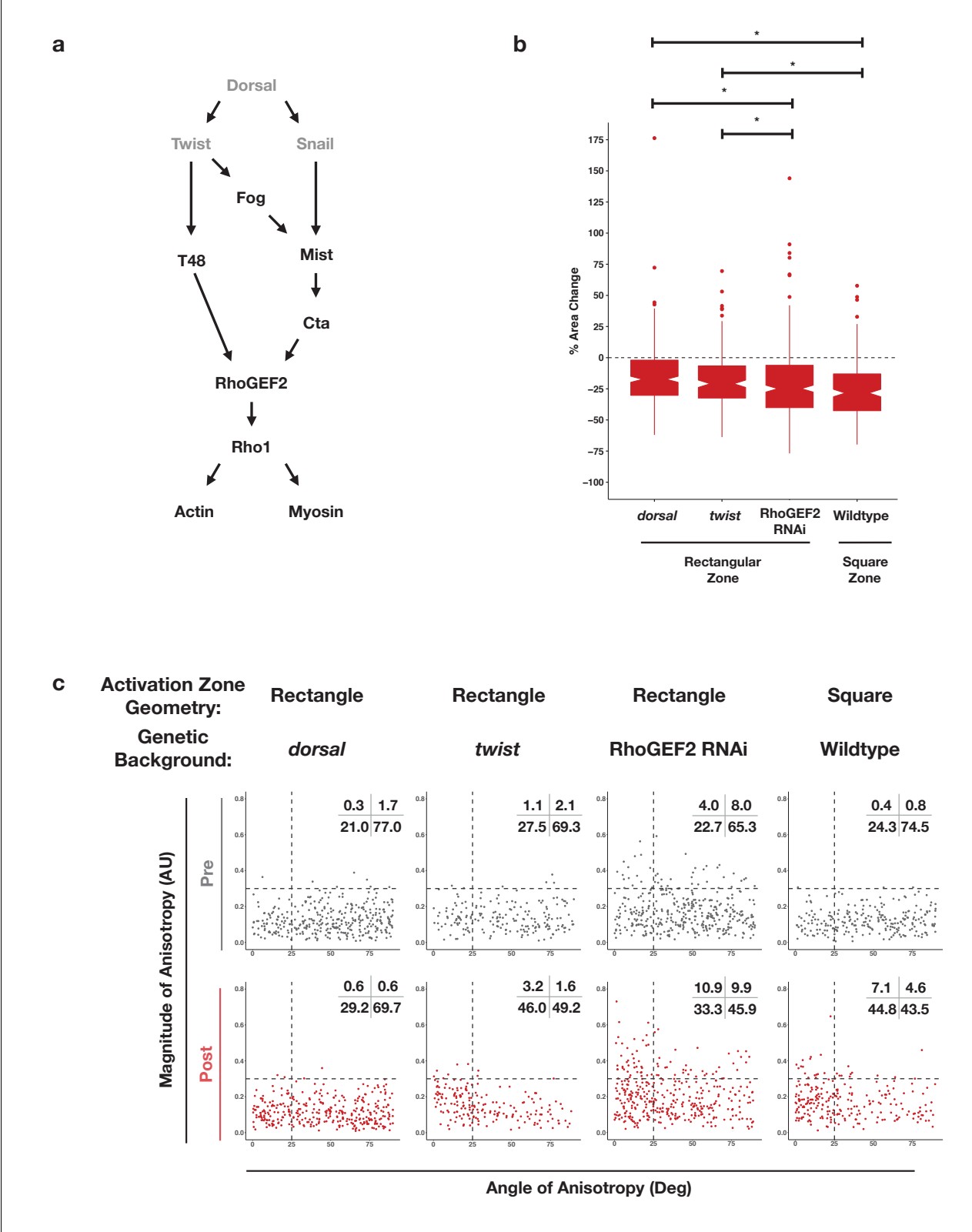

**Figure 5.** Dorsal is required for and Twist promotes aligned, anisotropic apical constriction in response to ectopic Rho1 activation. (a) Schematic of genetic logic for ventral cell identity. Gray text indicates transcription factors. (b) Quantification of apical area change induced by optogenetic Rho1 activation in wild-type and mutant backgrounds. Statistical analysis: Kruskal-Wallis test, Chi$^2$ = 37.3, df = 3, p = 4.0 × 10$^{-8}$; Pairwise comparisons with Wilcoxon signed rank text, using BH method to adjust for multiple comparisons. * indicates p< 0.05. (c) Anisotropy scatter plots, as in *Figure 2b*, for

*Figure 5 continued on next page*

*Figure 5 continued*

wild-type embryos subjected to a square region of ectopic Rho1 activation or specified mutant embryos subjected to a rectangular zone of activation. Insets show percentage of cells in each quadrant. Cells in the upper left quadrant exhibit highly aligned, anisotropic apical constriction. See *Figure 5—figure supplement 1* for images of mutant and wild-type embryos following optogenetic activation. See *Figure 4—figure supplement 1* for plots of the changes in anisotropy of individual cells. See *Figure 4—figure supplement 2* for statistical analysis of data in (c). A total of 343 cells from four *dorsal* embryos, 189 cells from three *twist* embryos, 375 from five RhoGEF2 depleted embryos, and 239 cells from five square zone embryos were analyzed.

The online version of this article includes the following figure supplement(s) for figure 5:

**Figure supplement 1.** Representative ventral epithelia of embryos expressing the optogenetic components and Gap43-Ch while lacking Dorsal or Twist protein, depleted of RhoGEF2, or subjected to a square zone of Rho1 activation.

Our experiments in the dorsal epithelium suggest that an asymmetric zone of Rho1 activation is not always sufficient to generate aligned, anisotropic apical constriction. However, an asymmetric zone of Rho1 activation might nevertheless contribute to the cell shape changes seen in ventral epithelial cells during ventral furrow formation. To address this question, we locally activated Rho1 in a square region in ventral cells before any obvious apical constriction. The area of the square activation zone was chosen so as to leave non-activated cells within the imaging field on either side of the activation zone. This resulted in the area of the square activation zone being ~ 2/3 the area of the rectangular activation zone. Square activation regions result in less highly anisotropic constrictions than rectangular activation regions (*Figure 5c* v. *Figure 4d* -VentralPost, *Figure 5—figure supplement 1*, *Figure 4—figure supplement 1*, *Figure 4—figure supplement 2*). Thus, even though an asymmetric zone of Rho1 activation is not sufficient to induce anisotropic apical constriction in the dorsal epithelium, the asymmetry of the zone of Rho1 activation contributes to the highly aligned anisotropic apical constriction in the ventral epithelium.

Taken together, these results suggest that an asymmetric zone of Rho1 activation and ventral-specific factors, genetically downstream of Dorsal and Twist, independently contribute to the ability of ventral cells to respond to ectopic Rho1 activation with aligned, anisotropic apical constriction.

## Acceleration and spreading of deformations within the endogenous ventral furrow region

Thus far, we have focused on the dorsal or ventral epithelium before the endogenous ventral furrow begins invaginating. We next asked whether optogenetic Rho1 activation would affect an already invaginating ventral furrow. Activation of Rho1 in a subset of cells locally accelerates their invagination (*Figure 6*, *Figure 6—video 1*). This suggests Rho1 activity is rate-limiting during the invagination of the endogenous ventral furrow. Notably, the invagination of neighboring ventral furrow cells, outside of the defined activation region is also accelerated (*Figure 6*, red arrows). These non-cell autonomous responses occur concomitantly with furrow acceleration, a time scale that is consistent with mechanical transmission of forces, rather than a mechanochemically-mediated response.

## Differential responses of cells flanking the Rho1 activation zone in the dorsal and ventral epithelium

Collectively, the results presented here suggest that ventral and dorsal cells exist in distinct environments and are exposed to distinct forces that impact their behavior. To further explore this possibility, we generated ectopic zones of Rho1 activation and focused on the behavior of cells adjacent to these zones. In the ventral epithelium, we observed extensive bending of non-activated cells towards optogenetically-induced furrows (*Figure 7b*, filled arrowheads). This bending is readily visualized in maximum intensity projections of the ventral surface post optogenetic activation, and it routinely extends several rows outside of the zone of photoactivation (*Figure 7b*, filled arrowheads). In contrast, long-range bending toward the ectopic furrows is not observed in the dorsal epithelium. Rather, the cells immediately adjacent to ectopic dorsal furrows exhibit substantial stretching of their apical surfaces (*Figure 7a*, open arrowheads). Notably, cells flanking ectopic furrows exhibit a similar extent of bending as is seen during endogenous ventral furrow formation (*Figure 2e*). However, these cells exhibit substantially less apical constriction than the cells that bend toward endogenous furrows, as this ectopic bending is induced precociously.

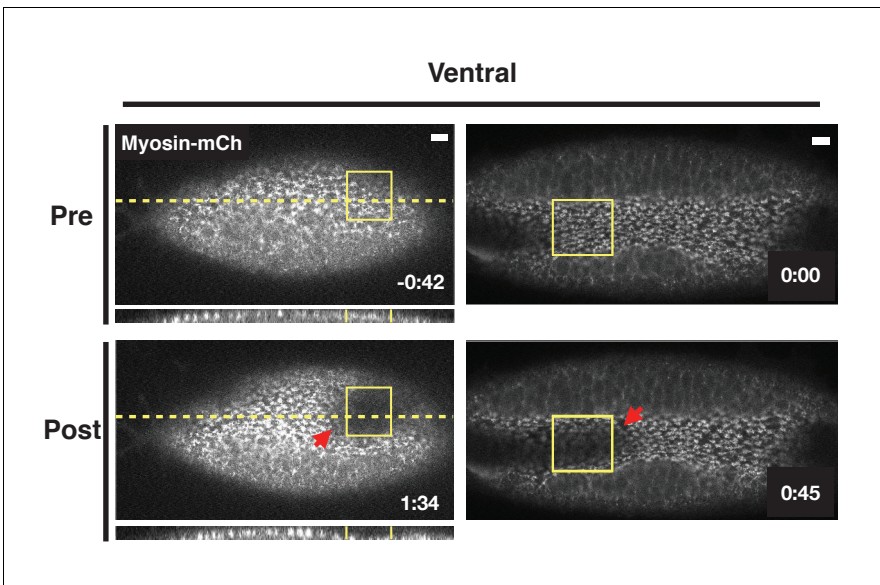

**Figure 6.** Optogenetic Rho1 activation accelerates the invagination of the endogenous ventral furrow. Ventral epithelia of two different embryos expressing the optogenetic components and Myosin-Ch and exhibiting an established furrow. Rho1 was activated in the yellow boxes. The acceleration of the invagination of the endogenous ventral furrow extends outside of the zone of optogenetic Rho1 activation (red arrows). Data representative of 3/5 embryos. Scale bars are 10 μm.

The online version of this article includes the following video for figure 6:

**Figure 6—video 1.** Movie of embryo shown in *Figure 6*, right panel.

https://elifesciences.org/articles/56893#fig6video1

We quantified the bending of non-activated cells toward ectopic furrows by measuring the change in the position of their apical centroids along the dorsal-ventral axis during the induction of the ectopic furrows, as in *Figure 2e*. Consistent with our visual observations, the centroids of the apical surface of non-activated ventral cells move substantially during the deformation of the photo-activated region, while the centroids of the apical surface of non-activated dorsal cells exhibit little movement during the comparable time (*Figure 7c*). Notably, dorsal cells neighboring ectopic furrows are strongly biased toward expanding their apical surfaces, while most ventral cells exhibit modest contraction of their apical surfaces. Thus, not only do activated ventral and dorsal cells respond distinctly to optogenetic activation of Rho1, but the non-activated ventral and dorsal cells neighboring these regions of ectopic Rho1 activation respond distinctly to ectopic furrows. These differential responses both within and adjacent to the activation zones suggest that cells within the two epithelia experience distinct forces that functionally modulate their behavior.

## Discussion

Experimental evidence and computational modeling suggest Rho1 activation could be the primary regulator of cell and tissue-level shape changes during ventral furrow formation. To test this model, we assessed whether an asymmetric zone of Rho1 activity is sufficient to induce this morphogenetic process in the *Drosophila* embryo. While optogenetic activation of Rho1 in the dorsal epithelium can induce a furrow, it does not recapitulate all cell- and tissue-level aspects of ventral furrow formation. However, Rho1 activation in the ventral epithelium induces cell- and tissue-level shape changes highly reminiscent of endogenous ventral furrow formation. Importantly, these changes are induced to occur earlier and more rapidly than they normally would. We propose that this context-dependent response to ectopic Rho1 activation arises because cells within dorsal and ventral epithelia experience distinct extrinsic forces and transmit their intracellular contractility over different length scales.

## A robust, ubiquitously expressed optogenetic probe enables transient activation of Rho1 in *Drosophila*

The LOV-domain-based optogenetic probe generated in this study is expressed ubiquitously throughout the *Drosophila* lifecycle. This ubiquitous and non-perturbing expression allows Rho1 activation to be readily controlled in any *Drosophila* tissue without the need to combine the probe with tissue-specific drivers. This probe acts rapidly, inducing Rho1 activity within a minute of photoactivation. Precise spatial control of Rho1 activation can be induced using a range of standard fluorescent imaging methods. Ectopic deformations induced by optogenetic Rho1 activation in the dorsal embryonic epithelium are limited to the zone of optogenetic Rho1 activation, and, in the wing peripodial epithelium, Rho1 can be activated with subcellular precision.

Using this optogenetic approach, we demonstrate that ectopic Rho1 activation is sufficient to induce ectopic, tissue-level shape changes throughout the embryonic epithelium just prior to and at the onset of gastrulation. The cell shape changes induced by optogenetic Rho1 activation in ventral cells closely mirror those seen during endogenous ventral furrow formation, and ectopic Rho1 activation can modulate endogenous ventral furrow formation. This suggests that the potency of optogenetic activation of Rho1 via the LOV probe is on par with endogenous Rho1 activation during ventral furrowing.

Deformations induced by optogenetic activation of Rho1 persist through the duration of optogenetic activation. However, the cells in these ectopic furrows rapidly revert to their pre-activation positions and expand their apical areas following cessation of photoactivation, concurrent with rapid dissipation of optogenetically induced myosin. Similar reversibilty occurs in the other tissues we examined as well as in cultured cells (*Wagner and Glotzer, 2016*; *Oakes et al., 2017*). This reveals the existence of potent, widespread mechanisms for inactivating Rho1 and its effectors. We infer that ventral furrow formation is driven by sustained Rho1 activation that overcomes this global inhibition.

## PR-GEF and RhoGEF2-CRY2 induce distinct cellular responses

Our results are partially consistent with previous work, which activated Rho1 via membrane recruitment of a light-responsive RhoGEF2-CRY2 fusion protein (*Izquierdo et al., 2018*). Both optogenetic systems induce ectopic deformations in the dorsal embryonic epithelium, but only RhoGEF2-CRY2 induces pulsatile Rho1 activity and anisotropic apical constriction in the dorsal epithelium.

The two systems use different RhoA/Rho1-specific GEFs, and this may underlie the differing results; LOV recruits LARG(DH), whereas CRY2 is fused to RhoGEF2(DHPH). Despite LARG being an extremely potent RhoA activator in vitro (*Jaiswal et al., 2013*), the transgene expressing LARG(DH) is well tolerated (*Table 1*), suggesting this recruitable GEF is non-perturbing. To directly compare LARG and RhoGEF2, we generated flies expressing SspB-GFP-RhoGEF2(DHPH) from the same genomic location as PR-GEF. SspB-GFP-RhoGEF2(DHPH) does not readily homozygose, even in the absence of the LOVSsrA membrane anchor (*Table 1*), suggesting it has significant light-independent activity. PH domains of the GEF subfamily that includes RhoGEF2 and LARG bind RhoA-GTP, and, in vitro, the interaction between the PH domain and membrane-bound RhoA-GTP potentiates GEF activity by up to 40-fold (*Chen et al., 2010*; *Medina et al., 2013*). Introducing two point mutations (F1044A, I1046E) into the PH domain of RhoGEF2, which are predicted to disrupt its binding to RhoA-GTP, allows the resultant transgene to readily homozygose (*Table 1*). These observations are consistent with RhoGEF2-CRY2 acting via a feedforward mechanism where it can be recruited by Rho1-GTP via its PH domain and thereby amplify Rho1-GTP. The ability of RhoGEF2-CRY2 to amplify both endogenous and light-induced Rho1 activity would be predicted to be particularly potent when it is overexpressed from a UAS promoter via Gal4. Feedforward activation via RhoGEF2-CRY2 may combine with the aforementioned mechanisms for Rho1 inactivation to generate the pulsatile Rho1 activity observed with RhoGEF2-CRY2 (*Izquierdo et al., 2018*). Amplification of endogenous Rho1 activity by RhoGEF2-CRY2 could also explain the anisotropic apical constrictions induced when this probe is optogenetically activated in the dorsal epithelium: Activated cells in this epithelium would need to deform against increased resistive forces exerted by their neighbors as a result of chronic Rho1 activation.

Although the GEF domain of RhoGEF2 is perturbing when over-expressed as an isolated domain, in the context of the full length protein, its ability to generate positive feedback via its PH domain

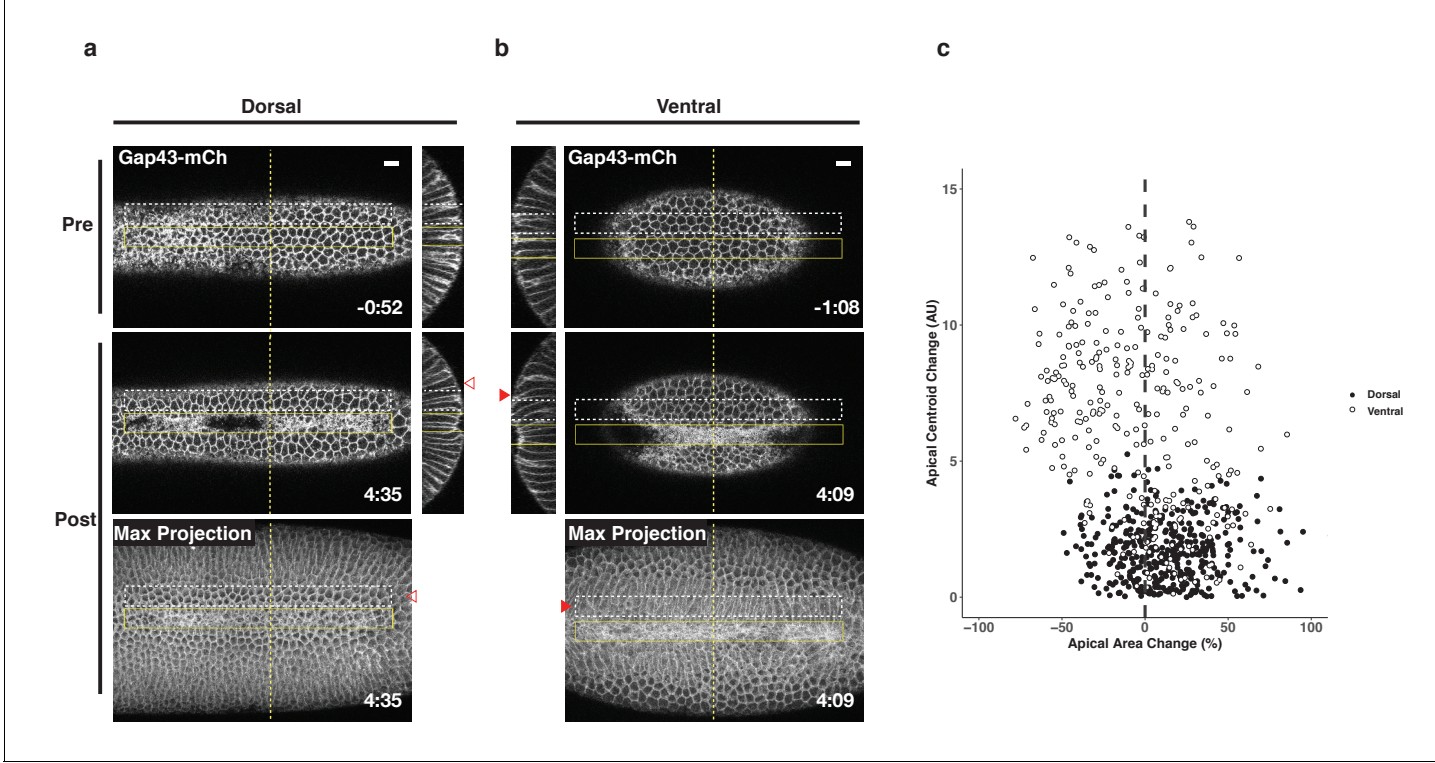

**Figure 7.** Non-activated cells bend towards ectopic furrows specifically in the ventral epithelium.  (**a–b**) Dorsal (**a**) and ventral (**b**) epithelium of embryos expressing the optogenetic components and Gap43-mCh before the onset of cell shape changes. Local Rho1 activation within the yellow box induces ectopic furrows in both the dorsal and ventral epithelium. Bottom panels: Maximum intensity projections of the indicated time point. Data representative of 4/4 dorsal and 4/4 ventral embryos. (**c**) Quantification of cell shape changes exhibited by cells neighboring ectopic furrows. X axis is the percent apical area change; Y axis is the change in position of the centroid of the apical cell surface along the dorsal-ventral axis. A total of 407 dorsal cells from four embryos and 298 ventral cells from four embryos were quantified. The white dashed boxes in a and b indicate the 'neighbor' cells that were quantified for these two embryos. Filled arrowheads indicate long-range bending of ventral cells; open arrowheads indicate the corresponding cells in the dorsal epithelium. Dorsal cells at the periphery are not perpendicular to the imaging plane before or after photoactivation and therefore do not appear as hexagons in the maximum intensity projection images. Time zero indicates the first pulse of blue light activation. Scale bars are 10 μm.

may contribute to the Rho1 activity pulses observed during ventral furrow formation (*Martin et al., 2009*; *Mason et al., 2016*).

## Endogenous and ectopic deformations involve a combination of intrinsic and extrinsic forces

The differences seen in the dorsal and ventral epithelium following optogenetic activation of Rho1 are not limited to the response of the activated cells. In the ventral, but not dorsal, epithelium several rows of cells bend toward ectopic furrows. Furthermore, optogenetic activation of Rho1 causes acceleration of the invagination of an already initiated ventral furrow that spreads rapidly, within

**Table 1.** Recruitable GEF viability tests.

| Cross scored | Possible genotype | Observed | Expected | Chi² |
|---|---|---|---|---|
| SspB-GFP-LARG(DH)/CyO x<br>SspB-GFP-LARG(DH)/CyO | 1) LARG (DH)/CyO<br>2) LARG(DH)/LARG(DH) | 379<br>165 | 362.7<br>181.3 | 2.198 |
| SspB-GFP-RhoGEF2(DHPH)/CyO x<br>SspB-GFP-RhoGEF2(DHPH)/CyO | 1) RhoGEF2(DHPH)/CyO<br>2) RhoGEF2(DHPH)/RhoGEF2(DHPH) | 318<br>54 | 248<br>124 | 59.27 |
| SspB-GFP-RhoGEF2(DHPH-F1044A,I1046E)/CyO x<br>SspB-GFP-RhoGEF2(DHPH-F1044A,I1046E)/CyO | 1) RhoGEF2(DHPH*)/CyO<br>2) RhoGEF2(DHPH*)/RhoGEF2(DHPH*) | 430<br>192 | 414.7<br>207.3 | 1.694 |

seconds, beyond the zone of optogenetic activation. These results suggest that optogenetically-induced intrinsic forces are readily transmitted over several cell diameters within the ventral epithelium.

Cells of the endogenous ventral furrow do not deform in isolation, rather constricting ventral cells exert forces on, and experience forces from, other cells within and outside of the ventral furrow [see (*Lye and Sanson, 2011*) for review]. This intercellular transmission of forces requires cell-cell coupling through junctional proteins, such as cadherins, junction-cytoskeletal linkages, such as those mediated by catenins, and the cytoskeleton itself. This set of molecular interactions, constitute a dynamic force transmission chain, that may be stabilized by force-mediated feedback (*Pinheiro and Bellaïche, 2018*). These interactions likely contribute to coordinated intracellular contractility throughout the ventral epithelium during endogenous ventral furrow formation and the bending of cells flanking endogenous or light-induced furrows.

## Requirements for ventral-specific responses to Rho1 activation

Despite the ability of asymmetric zones of Rho1 activation to induce deformations in both dorsal and ventral embryonic epithelia, only Rho1 activation in the ventral epithelium induced strong, aligned, anisotropic apical constriction of activated cells and bending of neighboring cells.

Results shown here indicate that Dorsal is required for and Twist promotes ventral-specific anisotropic apical constriction, consistent with the idea that difference(s) between the ventral and dorsal epithelia result from dorsal-ventral transcriptional patterning. Twist was shown previously to stabilize Rho1-driven apical constriction (*Martin et al., 2009*). Here, Twist promotes anisotropic apical constriction induced by sustained Rho1 activation. Although it is possible that these two defects result from loss of the expression of a single Twist target gene, Twist controls the expression of multiple genes, and several of these targets may make independent contributions to ventral furrow formation. As ventral cells lacking Twist exhibit weakly aligned, anisotropic apical constriction, a Twist-independent mechanism for generating aligned, anisotropic apical constriction must also exist; Snail is an excellent candidate for this role.

It is possible that Dorsal and/or Twist may be required for anisotropic apical constriction because each factor promotes Rho1 activation by RhoGEF2. However, results shown here suggest that elevated Rho1 activity alone does not explain this ventral-specific response. First, our experiments are performed prior to significant accumulation of endogenous myosin. Additionally, ventral cells depleted of RhoGEF2 exhibit an increase in magnitude and alignment of anisotropy following ectopic Rho1 activation, albeit to a lesser degree than wildtype ventral cells. Given the disorganization of the ventral epithelium following RhoGEF2 depletion, likely due to defects in cytoskeletal organization and cellularization (*Padash Barmchi et al., 2005*), it is notable that cells in this epithelium still exhibit an increase in aligned, anisotropic apical constriction following optogenetic activation of Rho1.

Previous work has identified several molecular and cellular differences between dorsal and ventral cells during early embryogenesis. For example, as cellularization completes and before ventral furrow formation initiates, junctional myosin is detected specifically in ventral cells (*Rauzi et al., 2015*). Adherens junctions and adjacent clusters of the polarity protein Bazooka disassemble from a subapical position by a Snail-dependent and myosin-independent mechanism in ventral, but not dorsal, cells (*Weng and Wieschaus, 2016*; *Kölsch et al., 2007*; *Weng and Wieschaus, 2017*). As ventral furrow formation progresses, Rho1 is activated in a subset of ventral cells, driving apical-medial accumulation of myosin and actin (*Rauzi et al., 2015*; *Dawes-Hoang et al., 2005*; *Fox and Peifer, 2007*; *Nikolaidou and Barrett, 2004*). In a myosin-dependent manner, dense clusters of adherens junction proteins accumulate apically (*Weng and Wieschaus, 2016*; *Kölsch et al., 2007*). Bazooka also reaccumulates apically shortly after the junctions reassemble (*Weng and Wieschaus, 2017*). Rap1 and its GEF Dzy are required for proper anisotropic apical constriction and ventral furrowing, and these factors may regulate these junctional rearrangements (*Sawyer et al., 2009*; *Spahn et al., 2012*). Ventral junctions also recruit the tension-sensitive protein Ajuba (*Winkelman et al., 2020*; *Sun et al., 2020*; *Rauskolb et al., 2019*). These results indicate that ventral cells undergo a number of changes that might alter their response to ectopic Rho1 activation. The experiments presented here directly test this prediction. Notably, of the aforementioned differences in dorsal and ventral cells, only Snail-dependent disassembly of adherens junctions and Bazooka occur before the onset of myosin-mediated contractility in the ventral epithelium. This Snail-dependent modulation of adherens junctions

and associated proteins is a prime candidate to cooperate with Twist to modulate the differential responses of dorsal and ventral cells to optogenetic activation of Rho1.

## Conclusion

The cellular behaviors observed during light-induced furrow formation in the ventral, but not dorsal, epithelium are remarkably similar to those that occur during endogenous ventral furrowing (*Costa et al., 1994*; *Leptin et al., 1992*; *Leptin and Grunewald, 1990*; *Sweeton et al., 1991*). These shape changes were widely thought to occur as a direct consequence of the transcriptional induction of Rho1-dependent contractility in the ventral epithelium. By comparing identical patterns and intensity of Rho1 activation in wild-type and mutant tissues, we have shown that dorsoventral patterning has additional relevant targets beyond Rho1 activation. We propose that ventral-specific behaviors arise from the expression of factors that modulate the cytoskeleton and its connection to adherens junctions as well as promote strong intercellular coupling among cells of the ventral epithelium.

# Materials and methods

## Plasmids

Plasmids used in this studied are listed in *Supplementary file 1*. pUbi-stop-mCD8GFP containing an attB site and pUbi >mEGFP-Anillin(RBD) were gifts from T. Lecuit. Plasmids created for this study were generated using SLiCE (*Zhang et al., 2012*) or one-step isothermal in vitro recombination (*Gibson et al., 2009*). Stargazin-GFP-LOVpep and PDZx2-mCherry-LARG(DH) plasmids were published previously (*Wagner and Glotzer, 2016*). Venus-iLID-CAAX and tgRFPt-SspB WT were obtained from Addgene (60411, 60415). pMT > Gal4 (*Klueg et al., 2002*) was obtained from the *Drosophila* Genomics Resource Center.

## Fly stocks

*Drosophila melanogaster* was cultured using standard techniques at 25°C. Both male and female animals were used. Stocks used in this study include *pSqh >Gap43-mCherry/TM3*, generated by P-element insertion and was a gift from A. Martin; *pSqh >Sqh* mCherry (*Martin et al., 2009*); Δ halo AJ *twist*[Ey53R12]/CyO, a gift from M. Leptin; *dl¹ cn¹ sca¹*/CyO (BID: 3236); *UAS > RhoGEF2 shRNA* (BID: 76255); *P(mat-tub-Gal4)mat67* (BID: 7062); *Sqh-GFP* (BID: 57144).

Transgenic flies were generated by PhiC31-directed integration (GenetiVision). Transgenic lines generated for this study include: *Ubi > Stargazin-GFP\*-LOVSsrA (attP2)*, *Ubi > Stargazin-GFP\*-LOV (I427V) SsrA (attP2)*, *Ubi > SspB-GFP-LARG(DH) (VK37)*, *Ubi > SspB-GFP-LARG(DH) (VK31)*, *Ubi > SspB-GFP-RhoGEF2 (DHPH) (VK37)*, *Ubi > SspB-GFP-RhoGEF2(DHPH-F1044A, I1046E) (VK37)*, *Ubi > SspB mScarlet (VK37)*, *Ubi > mCherry-Anillin(RBD) (attP40)*.

Genotypes of flies used in each experiment are listed in *Supplementary file 2* and *Supplementary file 3*.

## S2 cells

A total of 3.1 × 10⁶ S2 cells were transfected with 100 ng pMT >tagRFP SspB and 250 ng pMT >Stargazin-GFP\*-LOVSsrA or 250 ng pMT >Stargazin-GFP\*-LOV(I427V)SsrA using dimethyl-dioctadecyl-ammonium bromide (Sigma) (*Han, 1996*) at 250 μg/mL in six-well plates. Expression from the pMT promoter was induced 2 days after transfection by addition of 0.35 mM $CuSO_4$. Cells were imaged live 24 hr after $CuSO_4$ induction. Fifty μl of the S2 cell culture was plated on a glass slide and covered with a coverslip. Clay feet were used as spacers between the slide and coverslip. See *Supplementary file 5* for activation protocol details.

## Preparation of *Drosophila* tissues for live imaging

*Drosophila* embryos were collected on apple juice agar plates for 90 min and aged for 90–120 min at 25°C such that a majority of embryos were completing cellularization at the time of mounting. Embryos were dechorionated in 30% bleach for 1 min, rinsed in water, aligned on an apple juice agar pad, and mounted on a coverslip with embryo glue (adhesive from double sided tape dissolved in heptane). The imaged surface (dorsal or ventral) was mounted on the coverslip. This coverslip was

affixed via petroleum jelly to a metal slide with a hole in the center. Embryos were covered with halocarbon oil 200 immediately after mounting; they were not compressed.

Central nervous systems were dissected from wandering third instar larvae in Schneider's *Drosophila* Medium (Sigma) supplemented with 10% Fetal Bovine Serum (Thermo Fisher Scientific). Central nervous systems were imaged in a chamber comprising a coverslip affixed with petroleum jelly to a metal slide with a hole in the center. Following dissection, CNSs were mounted in the chamber such that their dorsal side contacted the coverslip. The chamber was flooded with Chan and Gehring's balanced solution (*Chan and Gehring, 1971*) to completely cover the CNS, and a gas-permeable membrane (YSI: 5793) was placed over the chamber to limit evaporation. These chambers were imaged on an inverted microscope.

Wing imaginal discs were dissected from wandering third instar larvae in S2 cell media supplemented with 10% FBS. Wing discs were mounted between a slide and glass coverslip in 50 µL Chan and Gehring's balanced solution. Clay feet were used as spacers between the slide and coverslip.

To prepare pupal nota, whole pupae were extracted from their pupal cases 18 hr post pupariation and mounted on a glass slide in a humid chamber, as described previously (*Zitserman and Roegiers, 2011*). Pupal nota were imaged on an upright microscope.

To image egg chambers, ovaries were dissected from 3- to 5-day-old females aged on yeast. Individual stage 10 egg chambers were isolated and mounted between a coverslip and a slide. Clay feet were used as a spacer between the slide and coverslip.

## Live imaging and optogenetic experiments

Global activation experiments were performed on a 63x/1.4 numerical aperture (NA) oil immersion lens on a Zeiss Axiovert 200M equipped with a Yokogawa CSU-10 spinning disk unit (McBain) and illuminated with 50 mW, 473 nm and 20 mW, 561 nm lasers (Cobolt) or on a Zeiss Axioimager M1 equipped with a Yokogawa CSU-X1 spinning disk unit (Solamere) and illuminated with 50 mW, 488 nm and 50 mW, 561 nm lasers (Coherent). Images were captured on a Cascade 1K electron microscope (EM) CCD camera, a Cascade 512BT (Photometrics), or a Prime 95B (Photometrics) controlled by MetaMorph (Molecular Devices). Photoactivation was accomplished by illuminating the sample with 488 nm light for the indicated exposure times (*Supplementary file 4* and *Supplementary file 5*).

Imaging of non-activated embryos and local activation experiments were performed on a inverted Zeiss LSM880 laser scanning confocal microscope with a 40X/1.4 numerical aperture (NA) objective. mCherry or mScarlet fluorescence was excited using the 561 nM solid state laser and was detected via a GaAsP spectral detector. Activation regions, indicated with yellow boxes throughout this manuscript, were defined in the 'Bleaching' module. Pixels within the defined activation zone were exposed to 488 nm light attenuated to 0.01 or 0.1 percent laser transmittance, using an Acousto-optic tunable filter, for 15 iterations every 20 s for the duration of the activation period. In general, we acquired a 'pre-activation' Z-Series of Gap43-mCh or Sqh-mCh, activated the defined region with 488 nm light in a single Z-plane, and acquired a 'post-activation' Z-Series of Gap43-mCh or Sqh-mCh. See *Supplementary file 4* and *Supplementary file 5* for specific activation protocols for each experiment.

## Image processing and cell shape analysis

All images were processed with FIJI (*Schindelin et al., 2012*). TissueAnalyzer (*Aigouy et al., 2010*), a FIJI plugin, was used to segment the embryonic epithelium and track cells for quantification of apical area, apical cell anisotropy, and apical cell centroid movement. 'Pre-' and 'Post-activation' Z-stacks were tracked separately in TissueAnalyzer (*Figure 4—figure supplement 4*), and data for the apical area, angle and magnitude of cellular anisotropy, and apical cell centroid were extracted from each timepoint and concatenated into a master data-base. Percent area change of the apical cell surface was calculated as (EndArea - StartArea)/StartArea*100. TissueAnalyzer computes a traceless symmetric tensor by integrating over the area of each cell (*Aigouy et al., 2010*). The components of this tensor are used to derive the magnitude and orientation of anisotropy. These parameters are normalized to cell area. As a consequence, the magnitude of anisotropy ranges from 0 to 1, with 0 being isotropic and one being maximally anisotropic.

To quantify myosin accumulation in the ventral epithelium over time for comparison to apical cell area, we first binned Z-stacks of Myosin-Ch $3 \times 3$ to account for the heterogeneity of myosin signal. We then generated a maximum intensity projection, defined a region of interest, and extracted the mean intensity value for that region.

To quantify myosin accumulation along the apicobasal axis of the embryo before and after photo-activation, we first thresholded images to avoid inclusion of the signal from cytoplasmic myosin. We then defined ten distinct 75 px by 60 px regions that spanned the horizontal length of the activated region. The ten regions were defined to account for the non uniform depth of deformations along the anterior-posterior axis. Myosin intensity was calculated as the raw integrated density of each thresholded region for each slice along the apicobasal axis. The values for myosin intensity at the apical most plane of all ten regions were than plotted as a mean and standard deviation. This was done iteratively for each subsequent position along the apicobasal axis until all positions were plotted.

Data were plotted in RStudio with ggplot2.

## Statistics

Kruskal-Wallis rank sum tests and Wilcoxon signed rank tests were used as noted. All statistical tests were performed in R Studio.

## Acknowledgements

This work was supported by R01GM085087, R35GM12709, and a France and Chicago Collaborating through the Sciences grant (MG), R01NS034783 (RGF), and NIH T32 GM007183 and NSF GRFP DGE-1144082; DGE-1746045 (AR). We thank Ed Munro, Angika Basant, and Yu Chung Tse for helpful comments on this manuscript. We thank Ed Munro, Sally Horne-Badovinac, Thomas Lecuit, MG lab members, and RGF lab members for helpful discussions and support. We thank Alyssa Harker for SQL training, Benoit Aigouy for TissueAnalyzer assistance, and Audrey Williams for help with oocyte dissections. We thank the Glick, Martin, Leptin, and Lecuit labs for generous sharing of reagents. We thank Ben Glick for access to the SnapGene molecular biology software (http://www.snapgene.com). Stocks obtained from the Bloomington *Drosophila* Stock Center (NIH P40OD018537) were used in this study. Reagents obtained from *Drosophila* Genomics Resource Center, supported by NIH grant 2P40OD010949 were used in this study.

## Additional information

### Funding

| Funder | Grant reference number | Author |
| --- | --- | --- |
| National Institute of General Medical Sciences | R35GM12709 | Michael Glotzer |
| University of Chicago | France And Chicago Collaborating in The Sciences | Michael Glotzer |
| National Institute of General Medical Sciences | R01GM085087 | Michael Glotzer |
| National Institute of Neurological Disorders and Stroke | R01NS034783 | Richard G Fehon |
| National Science Foundation | DGE-1144082 | Ashley Rich |
| National Science Foundation | DGE-1746045 | Ashley Rich |
| National Institutes of Health | T32GM007183 | Ashley Rich |

The funders had no role in study design, data collection and interpretation, or the decision to submit the work for publication.

## Author contributions
Ashley Rich, Conceptualization, Data curation, Software, Formal analysis, Validation, Investigation, Visualization, Methodology, Writing - original draft, Writing - review and editing; Richard G Fehon, Resources, Supervision, Writing - review and editing; Michael Glotzer, Conceptualization, Formal analysis, Supervision, Funding acquisition, Methodology, Writing - original draft, Project administration, Writing - review and editing

## Author ORCIDs
Ashley Rich (iD) https://orcid.org/0000-0003-2597-8104
Richard G Fehon (iD) http://orcid.org/0000-0003-4889-2602
Michael Glotzer (iD) https://orcid.org/0000-0002-8723-7232

## Decision letter and Author response
Decision letter https://doi.org/10.7554/eLife.56893.sa1
Author response https://doi.org/10.7554/eLife.56893.sa2

# Additional files

## Supplementary files
• Supplementary file 1. Plasmids used. References cited in this file: *Guntas et al., 2015*; *Munjal et al., 2015*; *Valbuena et al., 2020*; *Wagner and Glotzer, 2016*; *Wenzl et al., 2010*.

• Supplementary file 2. Genotypes and reproducibility: main figures.

• Supplementary file 3. Genotypes and reproducibility: figure supplements.

• Supplementary file 4. Activation protocols: main figures.

• Supplementary file 5. Activation protocols: figure supplements.

• Transparent reporting form

## Data availability
All data generated or analysed during this study are included in the manuscript and supporting files.

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
