## [Decision Letter]

**Acceptance summary:**

The spatial and temporal control of Myosin contractility is responsible for many cell and tissue shape changes, and nowhere is this manifest as in the case of mesodermal and endodermal invagination at the ventral and dorsal epithelium of a developing fly embryo, respectively. Using the power of *Drosophila* genetics and the ability to spatially and temporally activate contractility in the embryo, the authors demonstrate that the activation of myosin contractility that drives the formation of these important furrows of the developing embryo with distinct mechanistic consequences in the two regions.

**Decision letter after peer review:**

Thank you for submitting your article "Rho1 activation recapitulates early gastrulation events in the ventral, but not dorsal, epithelium of *Drosophila* embryos" for consideration by *eLife*. Your article has been reviewed by Utpal Banerjee as the Senior Editor, a Reviewing Editor, and two reviewers. The following individual involved in review of your submission has agreed to reveal their identity: Α Yap (Reviewer #3).

The reviewers have discussed the reviews with one another and the Reviewing Editor has drafted this decision to help you prepare a revised submission.

Summary:

The spatial and temporal control of Myosin contractility is responsible for many cell and tissue shape changes, and nowhere is this manifest as in the case of mesodermal and endodermal invagination at the ventral and dorsal epithelium of a developing embryo, respectively. Spatially and temporally orchestrated activation of Rho activation cause the activation of myosin contractility that drives the formation of these important furrows of the developing embryo.

At the cellular scale, apical myosin contractility is associated with apical cell constriction, tissue furrowing, and invagination. This manuscript adapt an existing optogenetic tool to spatio-temporally control myosin contractility in *Drosophila*. The strategy uses a membrane targeted LOV-ssrA peptide fusion and an exogenous RhoA GEF domain fused to the SspB protein, which will be recruited by ssrA upon photoactivation, similar to but modified from earlier studies (Izquierdo et al., 2018; Krueger et al., 2018). The authors refine this tool to make it more responsive to levels of induction as well as reversibility. They activate RhoA and induce contractility on both the dorsal and ventral side of the embryo, which results in furrows at both locations.

They find that ventral cells have a greater propensity to constrict and do so anisotropically. They emphasize the importance of genetic context by showing that these regional differences are substantially reduced in Twist or Dorsal mutants. This might be thought to be a logical finding (given the role of these genes in ventral furrow formation), but it is valuable to have it tested directly. In addition, the authors show that there is more of a response from flanking cells to constriction on the ventral side, again suggesting a ventral-dorsal difference in behaviour.

However, all the reviewers had serious reservations about the origin of difference between the material properties of the two tissues. Specific points about the differences in the material properties of the doral and ventral sides relate to the involvement of active responses from the perturbations used (points 1, 2) as well as the origin of these differences (3-5). If this manuscript may be considered for a revised submission, the issues are listed below need to be addressed.

Essential revisions:

1) The authors suggest that a key to this region specificity might lie in the mechanics of the two tissues. This is a plausible hypothesis, but one that is not explored further, and that rather limits the depth of the current manuscript. One striking observation is that ventral epithelial cells constrict their apical surfaces even outside the region of illumination, but dorsal cells do not. This suggests that there might be some active response at a distance that occurs specifically in the ventral epithelium. If so, this might give a clue to suggest that differences lie in active (rather than passive) mechanical responses of these two regions of the embryo – such as might arise from mechanosensitivity or mechanotransduction.

The authors could consider this with two sets of experiments:

a) Is Myosin II recruited to the cortex outside these regions of illumination; and is this different in ventral vs dorsal cells? (Here it will be important for the authors to exclude the possibility that this is an artefact of the experiment, as earlier they show that high illumination energies can activate myosin outside the illumination zone – at least in the larval wing peripodial epithelium.)

b) Is this regional difference affected in Dorsal or Twist embryos?

2) In Figure 2, the authors are inducing their optogenetic reagent while the endogenous pathway is apparently getting activated. That apical myosin is already accumulating during optogenetic stimulation is concerning because this endogenous activity will affect the measured response. The authors need to explicitly say which movies used during analysis had the endogenous RhoA response occurring. An alternative explanation for the differences between ventral and dorsal cells observed is that the normal ventral cell response adds to the optogenetic activation to give a different response. Conversely, the dorsal cells get stretched when ventral cells invaginate (Rauzi et al., 2015), which could lessen their apical constriction response. The timing of optogenetic activation on the dorsal side could therefore be critical. The authors should indicate the exact embryonic stage that the optogenetic activation was performed so that it is clear what other morphogenetic movements should be considered in the interpretation.

3) Another reason why dorsal cells may furrow with less apical constriction is that they are constricting via a different mechanism, such as by lateral shortening. Lateral shortening has been described to occur in the dorsal folds (Takeda et al. 2018). In Figure 2—figure supplement 1, myosin is enriched at the apical surface of ventral cells, but myosin looks less apically restricted in dorsal cells. The authors should quantify the amount of apical vs. lateral myosin accumulation to determine if the level and localization of RhoA activation is the same in each cell type. This is important to clarify as it also affects the interpretation of later results (i.e. apical constriction amount and anisotropy).

It is also important to note that the confocal excitation used is likely to activate RhoA broadly along the optical axis (i.e., apical-basal axis). Showing a cross-sectional view of the whole depth of the epithelial layer to show whether cell height is constant and whether activation is confined to the apical surface in dorsal and ventral cells would be ideal. One concern is that different behaviors are due to different levels of basal myosin, which may be activated more in dorsal cells or the effects seen may be a consequence of the depth of confocal excitation convoluted with differential expression of Myosin? This should be clearly distinguished.

4) The authors also argue that the Dorsal and Ventral epithelium may have different mechanical properties because “ventral-specific response occurs in less than a minute, a time scale that is consistent with mechanical, rather than mechanochemical, transmission of forces. Thus, cells within the ventral and dorsal epithelia may exhibit differential mechanical properties.”

Notably in Figure 3C,D: The authors argue that dorsal and ventral epithelia have different material properties, however, this assumes a uniform contractile response in the different tissues. The authors never compare the amount of myosin that is induced between ventral and dorsal cells, making it unclear whether the activation is the same. Furthermore, when comparing anisotropy, the analysis is not normalized for constriction. In addition, it is possible that the cytoskeletal networks equally are not attached to adherens junctions to the same extent. Without standardizing the input force and follow up experiments examining tissue prestress, it is impossible to compare the material properties, or make such defining statements.

5) They also show that activation of a rectangular region generates more anisotropic constriction than a square region in the ventral epithelium, but the rectangular region is not sufficient to induce an anisotropic constriction in the dorsal epithelium, again pointing to a material difference (Figure 3C versus Figure 4). However, they do not provide a clear description of how large was the square region and if this region were of the same area as the rectangle or a square with the same dimension as the long or small side of the rectangle. It would be important to examine contractility of the cells in the same regions where the rectangle is created, rather than over an entire square segment where myosin is also recruited, post RhoA activation. This relates to the experiment where they activate RhoA in an epithelium that is depleted of RhoGEF (Figure 4C). Here apical constrictions in the rectangle are even more anisotropic. Conversely, if RhoA were broadly activated throughout the epithelium, it is likely that the tissue would lose its anisotropy. Using the properties of the optogenetic tool, the authors have the capacity to test this, if they wish to claim differences in material properties. Further, they do not test the extent of anisotropy in the dorsal epithelium after depleting RhoGEF. It is likely that these data already exist with the authors and should be reported here.

6) Although the authors' study is a nice demonstration of using optogenetics to test cells' predisposition to apical constriction, which was not done before and will inform future optogenetic experiments at this stage. Classic and recent studies have demonstrated different behaviors in dorsal vs. ventral cells of the *Drosophila* embryo, including different organization in adherens junctions, actin levels, basal myosin, microtubules, and polarity (Sweeton et al., 1991; Koelsch et al., 2007; Wang et al., 2012; Polyakov et al., 2014; Rauzi et al., 2015; Jodoin et al., 2015; Weng and Wieschaus, 2016; Weng and Wieschaus, 2017; Takeda et al., 2018). It is unclear how the authors' current results connect to these past studies and whether the authors are uncovering something that is fundamentally new. There’s no need for new experiments to justify these arguments, but a slightly broader review of possible differences between dorsal and ventral cells in the Discussion would make this better appreciated.

---

## [Author Response]

Essential revisions:1) The authors suggest that a key to this region specificity might lie in the mechanics of the two tissues. This is a plausible hypothesis, but one that is not explored further, and that rather limits the depth of the current manuscript. One striking observation is that ventral epithelial cells constrict their apical surfaces even outside the region of illumination, but dorsal cells do not. This suggests that there might be some active response at a distance that occurs specifically in the ventral epithelium. If so, this might give a clue to suggest that differences lie in active (rather than passive) mechanical responses of these two regions of the embryo – such as might arise from mechanosensitivity or mechanotransduction.The authors could consider this with two sets of experiments:a) Is Myosin II recruited to the cortex outside these regions of illumination; and is this different in ventral vs dorsal cells? (Here it will be important for the authors to exclude the possibility that this is an artefact of the experiment, as earlier they show that high illumination energies can activate myosin outside the illumination zone – at least in the larval wing peripodial epithelium.)b) Is this regional difference affected in Dorsal or Twist embryos?

We agree with the reviewers that the molecular mechanism underlying the differential responses of dorsal and ventral cells to optogenetic activation of Rho1 remains an unanswered question in our manuscript, though we suggest our data narrows the possibilities. To test whether the differential responses of dorsal and ventral cells arise from an active response to optogenetic Rho1 activation at a distance in the ventral, but not dorsal, epithelium, we optogenetically activated Rho1 in the ventral epithelium and assayed whether myosin accumulates outside of the defined activation region. In contrast to the activation patterns used in most of the manuscript, we activated Rho1 along the dorsal-ventral axis, perpendicular to the eventual ventral furrow, to ensure that we could compare the response of activated and non-activated ventral cells at identical positions along the dorsal-ventral axis. Optogenetically-induced Myosin-Ch accumulation was strictly limited to the defined region of activation, suggesting that optogenetically activated Rho1 does not induce active responses at a distance in the ventral epithelium and rules out this mechanism as an explanation of the observed differential responses of dorsal and ventral cells to optogenetic Rho1 activation. These results are presented in Figure 3—figure supplement 1. While high light doses can result in diffuse activation, we strictly limited the laser dose in these experiments to prevent this from happening (the laser power was limited to 0.01% transmittance).

The quantitative analysis of anisotropy in *dorsal* and *twist* embryos suggest that these genes promote the ventral specific response (Figure 5). Furthermore, visual inspection of the cells neighboring optogenetic-induced deformations in embryos lacking Dorsal or Twist protein indicates there is no long-range bending of cells in the ventral epithelia of these mutants (see maximum intensity projections in Author response image 1 and compare YZ projections in Figure 4B and Figure 5—figure supplement 1).

**Author response image 1. sa2fig1:** Maximum intensity projections of ventral epithelia of embryos lacking the Dorsal (left) or Twist (right) protein and expressing the optogenetic components and Gap43-Ch. Rho1 was optogenetically activated in the yellow box.

2) In Figure 2, the authors are inducing their optogenetic reagent while the endogenous pathway is apparently getting activated. That apical myosin is already accumulating during optogenetic stimulation is concerning because this endogenous activity will affect the measured response. The authors need to explicitly say which movies used during analysis had the endogenous RhoA response occurring. An alternative explanation for the differences between ventral and dorsal cells observed is that the normal ventral cell response adds to the optogenetic activation to give a different response. Conversely, the dorsal cells get stretched when ventral cells invaginate (Rauzi et al., 2015), which could lessen their apical constriction response. The timing of optogenetic activation on the dorsal side could therefore be critical. The authors should indicate the exact embryonic stage that the optogenetic activation was performed so that it is clear what other morphogenetic movements should be considered in the interpretation.

To clarify when the optogenetic activations were performed relative to onset of ventral furrow formation, we generated a quantitative timeline of cell shape changes during ventral furrow formation in embryos that expressed the optogenetic components and Gap43-Ch but were never exposed to activating blue light (Figure 2). We also imaged embryos co-expressing Gap43-Ch and Myosin-GFP (and lacking the optogenetic components) to determine when Myosin accumulates relative to the start of apical constriction during ventral furrow formation (Figure 2—figure supplement 1). These data show that myosin begins to detectably accumulate, at most, two minutes prior to the onset of apical constriction. Thus embryos that lack detectable myosin have not initiated cell shape changes and embryos that do not exhibit cell shape changes accumulate little if any myosin.

We emphasize that the presence of the LOV domain-based probe, which is activated by blue light, precludes the simultaneous use of Gap43-Ch, Myosin-GFP, and photoactivation. Thus, in the context of the optogenetic experiments, cell shape changes and myosin accumulation were necessarily analyzed in parallel. Nevertheless, we believe the addition of the above discussed timeline of endogenous ventral furrow formation helps to clarify the timing of our optogenetic experiments relative to early embryonic development.

In activation experiments in which we assayed cell shape in ventral cells with Gap43Ch, a comparison of the magnitude and variance of cell areas indicate that our optogenetic activations begin substantially before cell area changes (Figure 2C,D, pre-activation) and before signs of cellular anisotropy appear (cf. Figure 4D vs Figure 2B).

In activation experiments in which we assayed Myosin-Ch in ventral epithelia, we could directly assess when endogenous myosin accumulated. As the optogenetic zone of activation is somewhat narrower than the endogenous zone, we could readily identify if an embryo was initiating furrow formation before the activation protocol and we did not include such embryos in our analysis. Notably, in some cases, the optogenetically-induced furrow is displaced from the eventual ventral midline, resulting in the induced furrow being resolved from the endogenous furrow in both space and time (Figure 3B).

For experiments in dorsal epithelia, photoactivation was initiated before cephalic fold formation; a short delay separates the initiation of the ventral furrow and initiation of this fold. Our activations would be completed before the stretching of dorsal cells, which begins about 10 minutes after ventral furrow initiation (Rauzi, et al., 2015).

Additionally, in subsection “Rho1 activation is sufficient to induce reversible furrows in the *Drosophila* embryonic epithelium”, we provide several lines of evidence that support the argument that the tissue-level shape changes and myosin accumulation observed in ventral epithelia are light-dependent.

3) Another reason why dorsal cells may furrow with less apical constriction is that they are constricting via a different mechanism, such as by lateral shortening. Lateral shortening has been described to occur in the dorsal folds (Takeda et al. 2018). In Figure 2—figure supplement 1, myosin is enriched at the apical surface of ventral cells, but myosin looks less apically restricted in dorsal cells. The authors should quantify the amount of apical vs. lateral myosin accumulation to determine if the level and localization of RhoA activation is the same in each cell type. This is important to clarify as it also affects the interpretation of later results (i.e. apical constriction amount and anisotropy).It is also important to note that the confocal excitation used is likely to activate RhoA broadly along the optical axis (i.e., apical-basal axis). Showing a cross-sectional view of the whole depth of the epithelial layer to show whether cell height is constant and whether activation is confined to the apical surface in dorsal and ventral cells would be ideal. One concern is that different behaviors are due to different levels of basal myosin, which may be activated more in dorsal cells or the effects seen may be a consequence of the depth of confocal excitation convoluted with differential expression of Myosin? This should be clearly distinguished.4) The authors also argue that the Dorsal and Ventral epithelium may have different mechanical properties because “ventral-specific response occurs in less than a minute, a time scale that is consistent with mechanical, rather than mechanochemical, transmission of forces. Thus, cells within the ventral and dorsal epithelia may exhibit differential mechanical properties.”Notably in Figure 3C,D: The authors argue that dorsal and ventral epithelia have different material properties, however, this assumes a uniform contractile response in the different tissues. The authors never compare the amount of myosin that is induced between ventral and dorsal cells, making it unclear whether the activation is the same.

To determine where, along the apicobasal axis of cells, myosin accumulates following optogenetic activation of Rho1, we photoactivated Rho1 in embryos expressing the optogenetic components and Myosin-Ch. To ensure that Myosin-Ch accumulation was light-dependent we depleted RhoGEF2 from these embryos. We quantified the average intensity of myosin-Ch at 1 micron intervals along the apicobasal axis of the embryo before and after photoactivation (see Figure 3—figure supplement 3). Myosin accumulation is limited to the apical-most 3-4 microns of both dorsal and ventral epithelia; which is reasonable given the dimensions of a confocal point spread function in Z and the high sensitivity of the LOV domain. These results suggest that the optogenetic input is similar in the dorsal and ventral epithelium. Furthermore, the apically restricted localization of light-induced Myosin-Ch in both dorsal and ventral epithelia is inconsistent with lateral shortening occurring preferentially in either epithelium.

Furthermore, when comparing anisotropy, the analysis is not normalized for constriction.

We used TissueAnalyzer, described in Aigouy et al., 2010, to measure cell elongation, which served as a proxy for anisotropy. This measurement of cell elongation does normalize for cell area, as stated in the Supplemental Theoretical Procedures (Aigouy et al., 2010).

In addition, it is possible that the cytoskeletal networks equally are not attached to adherens junctions to the same extent. Without standardizing the input force and follow up experiments examining tissue prestress, it is impossible to compare the material properties, or make such defining statements.

As mentioned above we removed references to the material properties of the tissue.

5) They also show that activation of a rectangular region generates more anisotropic constriction than a square region in the ventral epithelium, but the rectangular region is not sufficient to induce an anisotropic constriction in the dorsal epithelium, again pointing to a material difference (Figure 3C versus Figure 4). However, they do not provide a clear description of how large was the square region and if this region were of the same area as the rectangle or a square with the same dimension as the long or small side of the rectangle.

We have clarified the size of the square activation zone in two ways: (1) We now show an example of a square activation zone in Figure 5—figure supplement1. We also state in the manuscript that the square activation zone is ~2/3 the area of the rectangular activation zone (subsection “Genetic requirements for ventral-specific responses”).

To be more specific, the rectangular zones of activation, such as those used in Figure 4 and Figure 5, were 914px long by 66px high, covering a total area of 60,423px2. The square zones of activation, such as those used in Figure 5, were 200px long by 200px high, covering a total area of 40,000px2. The dimensions of both the rectangular and square activation zones were chosen to leave visible cells bordering the activation zone so that the extent of ectopic invagination could be readily monitored in real time. We acknowledge that area, length, and height were not identical in these experiments, but we point out that a substantial number of cells remain activated by the square activation zone. Thus, we think the size of the square activation zone is sufficiently large to reveal cell and tissue level behaviors.

It would be important to examine contractility of the cells in the same regions where the rectangle is created, rather than over an entire square segment where myosin is also recruited, post RhoA activation.

If we understand this comment correctly, the reviewers are suggesting that the analysis should be restricted to the cells that lie in the region that is shared between the “standard rectangular activation region” and the “square activation region”. Presumably, the underlying concern is that by analyzing the entire square region we obscure a difference in the response of cells in the “shared region”. There are two reasons why we decided not to follow this suggestion. First, to restrict the analysis to cells in the “standard rectangular activation region”, we would need to arbitrarily specify a rectangular region as there are no fiduciary marks that would enable this specification. Second, and more importantly, there are no systematic differences in responses of the cells in the square activation region.

This relates to the experiment where they activate RhoA in an epithelium that is depleted of RhoGEF (Figure 4C). Here apical constrictions in the rectangle are even more anisotropic.

Our data do not show that optogenetically-induced apical constructions are more anisotropic in RhoGEF2 depleted epithelia than in wildtype epithelia. For example, 27.4% of activated wildtype cells exhibit highly aligned, anisotropic apical constriction (Figure 4D) while only 10.9% of activated cells depleted of RhoGEF2 exhibit highly aligned, anisotropic apical constriction (Figure 5C).

Conversely, if RhoA were broadly activated throughout the epithelium, it is likely that the tissue would lose its anisotropy. Using the properties of the optogenetic tool, the authors have the capacity to test this, if they wish to claim differences in material properties.

As mentioned above we removed references to the material properties of the tissue. This is an interesting suggestion, but we have not had the opportunity to perform this experiment.

Further, they do not test the extent of anisotropy in the dorsal epithelium after depleting RhoGEF. It is likely that these data already exist with the authors and should be reported here.

These data are now shown in Figure 4—figure supplement 1 and Figure 4—figure supplement 2.

6) Although the authors' study is a nice demonstration of using optogenetics to test cells' predisposition to apical constriction, which was not done before and will inform future optogenetic experiments at this stage. Classic and recent studies have demonstrated different behaviors in dorsal vs. ventral cells of the *Drosophila* embryo, including different organization in adherens junctions, actin levels, basal myosin, microtubules, and polarity (Sweeton et al., 1991; Koelsch et al., 2007; Wang et al., 2012; Polyakov et al., 2014; Rauzi et al., 2015; Jodoin et al., 2015; Weng and Wieschaus, 2016; Weng and Wieschaus, 2017; Takeda et al., 2018). It is unclear how the authors' current results connect to these past studies and whether the authors are uncovering something that is fundamentally new. There’s no need for new experiments to justify these arguments, but a slightly broader review of possible differences between dorsal and ventral cells in the Discussion would make this better appreciated.

We revised the Discussion to reflect that the differential responses of dorsal and ventral cells to optogenetic Rho1 activation may indeed result from previously identified cellular and molecular differences between these two populations of cells.